# Quantifying and Communicating Uncertain Climate Change Hazards in Participatory Climate Change Adaptation Processes

Laura Müller[1] and Petra Döll[1, 2]

[1] Institute of Physical Geography, Goethe University Frankfurt, Frankfurt, 60438, Germany.

[2] Senckenberg Leibniz Biodiversity and Climate Research Centre (SBiK-F) Frankfurt, Frankfurt, 60325, Germany.

*Correspondence to*: Laura Müller (la.mueller@em.uni-frankfurt.de)

**Abstract.** Participatory processes for identifying local climate change adaptation measures have to be performed all around the globe. As these processes require information about context-specific climate change hazards, knowledge about how to quantify climate change hazards and how to best communicate the potential hazards with their uncertainties is essential. In a

10 participatory process on water-related adaptation in a biosphere reserve in Germany, we used the freely available output of a multi-model ensemble provided by the ISIMIP initiative, which provides global coverage, to quantify the wide range of potential future changes in (ground)water resources. Our approach for quantifying the range of potential climate change hazards can be applied worldwide for local to regional study areas, and also for adaptation in agriculture, forestry, fisheries, and biodiversity. To support participatory climate change adaptation processes, we propose to communicate uncertain local

climate change hazards with percentile boxes rather than with boxplots or simple averages with the model agreement on the sign of change. This helps the stakeholders identify the future changes they wish to adapt to, depending on the problem (e.g., resource scarcity vs. resource excess) and their risk aversion. Using or adapting our quantification and communication approach, flexible climate change risk management strategies can and should be developed worldwide in a participatory and transdisciplinary manner involving stakeholders and scientists.

**Plain Language Summary.** All around the world, it is necessary to adapt to climate change, and people need to work together in local participatory processes to be able to identify the best local adaptation measures. Any development of adaptation measures requires information about the changes thay may occur in the future, for example changes in water resources or crop yield. As the future cannot be reliably predicted, a range of possible future changes should be considered. These can be

quantified with free data of global coverage from multiple computer models, which is available for many sectors like water, agriculture, forestry, fisheries, and biodiversity. In this paper, we show how to quantify the ranges of possible future changes in water resources using free global data and how to communicate them to stakeholders who want to identify adaptation measures. To optimize communication, we propose using "percentile boxes" instead of boxplots or simple averages with the model agreement on whether there will be an increase or a decrease in water resources. This way, people can better understand

what may happen in the future and decide what possible future they want to adapt to, for example to much less or to somewhat less water than today, depending on how much risk they are willing to take. Our quantification and communication approach

can support climate change adaptation processes worldwide, where stakeholders and scientists collaborate to develop flexible strategies for reducing climate change risks.

## 1 Introduction

Climate change alters environmental systems. Any alteration is likely to be a hazard as humans and other biota are adapted to unaltered conditions and future alterations are unknown. In combination with the vulnerability of the system under consideration, a climate change-related physical hazard leads to a risk that should be reduced by adaptation to climate change. As an example, water flows and storages on the continents are changing, and their temporal patterns cannot be assumed to be stationary anymore (Milly et al., 2008). This hazard makes water management, which aims at reducing risks for, e.g., human

water supply, more difficult as it is based on the evaluation of historic data and is generally adjusted to past conditions (Riedel and Weber, 2020). It is widely accepted that the management of environmental systems all around the globe has to be adapted to the changing climate and that robust and flexible climate change risk management strategies should be developed jointly by stakeholders and scientists in a participatory and transdisciplinary manner (Daniels et al., 2020; Döll and Romero-Lankao, 2017; Krueger et al., 2016; Scrieciu et al., 2021; Strasser et al., 2014).

To adapt the current management, stakeholders need to know how the system under consideration may develop in response to climate change. Given that adaptation to climate change impacts always pertains to the future, uncertainty is unavoidable (Lux and Burkhart, 2023). While some effects of climate change such as an increase in temperature or an increased precipitation variability are qualitatively well known, the quantification of changes, in particular for areas with a small spatial extent, is highly uncertain due to the complexity of the Earth system as well as the unpredictability of future greenhouse gas

emissions (Döll et al., 2015). Uncertainty means that we have limited knowledge about something (Marchau et al., 2019), i.e., "[it] is the inability to determine the true magnitude or form of variables or characteristics of a system […]" (Mahmoud et al., 2009, p. 806). This is why (a) future changes should be assessed with their uncertainty and (b) a suitable visualization should be found with which the future changes with their uncertainty can be communicated.

          Future changes are quantified with climate and impact models (Appendix A2) but only with a rather large degree of

uncertainty. Assuming the same greenhouse gas emissions scenarios, different climate models compute rather different future changes in climatic variables such as near-surface temperature and, even more so, precipitation, and different impact models will translate the same time series of climatic input variables into rather different time series of, e.g., groundwater recharge. It is state of the art to estimate the uncertainty of future changes by analyzing the output of multi-model ensembles (Döll et al., 2015). Multi-model ensemble output is the output of multiple impact models where each of the impact models is run various

60    times driven by the output of multiple climate models. The output of each climate model-impact model combination can be considered to be equally likely as it is, in most cases, impossible to say which of the models, each with different model algorithms and input data, is a better representation of reality than the other models. Thus, for each future change, the distribution of the changes simulated by all model combinations can be considered to represent a probability distribution. Then,

multi-model ensembles can be used to quantify the uncertainty of future changes caused by the uncertainties of climate and impact models, and these uncertainties can be classified as "shallow" (Döll et al., 2015; Döll and Romero-Lankao, 2017). In contrast, pathways of future greenhouse gas emissions are characterized by "deep" uncertainties such that their occurrence cannot be described by probabilities, and it is not even possible to rank them by their likelihood (Döll and Romero-Lankao, 2017). This level of uncertainty can be addressed by generating scenarios of alternative plausible futures. Therefore, for most evaluations, it is preferable to analyze separate multi-model ensembles for each emissions scenario. To confront various origins of uncertainties when modelling the future, Maier et al. (2016) suggested that the quantification of potential future changes should be based on the combination of three complementary paradigms, which are "a) anticipating the future based on best available knowledge, b) quantifying future uncertainty, [and] c) exploring multiple plausible futures" (Maier et al., 2016, p. 155, Fig. 1). Thus, it is essential to characterize the uncertainty of simulated changes that is caused by the multiple plausible future greenhouse gas emissions and the uncertainties of the applied climate and impact models (Riedel and Weber, 2020).

The Inter-Sectoral Impact Model Intercomparison Project (ISIMIP, www.isimip.org) provides freely available multi-model ensembles of many model output variables that are of interest to quantify climate change hazards in several impact sectors (water, lakes, biomes, forests, permafrost, agriculture (crop modelling), energy, health, coastal systems, fisheries and marine ecosystems, and terrestrial biodiversity; ISIMIP, 2019). The available impact model outputs mostly cover all land areas of the globe. For each impact variable, ISIMIP2b provides a time series for historic and future periods and several greenhouse gas eissions scenarios, which were computed by multiple global impact models (Frieler et al., 2017), with which the uncertainties of future changes in impacted variables can be characterized.

To support stakeholders who are responsible for identifying and performing robust and flexible local climate change risk management, experts need to translate the multi-model ensemble output into meaningful and usable information (Daniels et al., 2020). Van der Bles et al. (2019) developed a communication framework for epistemic uncertainty that "addresses who communicates what, in what form, to whom and to what effect while acknowledging the relevant context as part of the characteristics of the audience" (van der Bles et al., 2019, p. 3). Of the three formats to communicate uncertainty, numerical and visual formats convey a higher precision of uncertainty, while a low precision uncertainty is typically communicated verbally (van der Bles et al., 2019). Limited empirical evidence exists on which communication formats are more suitable, mostly on how verbal expressions are interpreted but very little on visual and numerical formats (van der Bles et al., 2019). While a wide variety of approaches, in particular visualization approaches, has been developed to communicate uncertainty, the most suitable format of uncertainty communication depends on the communicators' objectives, communication context, and audience, and no general recommendations for the perfect communication format in a certain context can be given (Spiegelhalter et al., 2011).

To communicate the quantified potential changes with their uncertainties visually, a suitable visualization format is needed and should not be translated into median or mean changes only, as this would suppress information about the actual uncertainty range of projected changes. Suitable ways to communicate and assess the potential range of future developments and thus identify the consequent adaptation needs should be identified, e.g., by analysing risks under a discrete number of

likely future developments of the physical system (Jack et al., 2020). Crosbie et al (2013) suggested showing the distribution of future changes in groundwater recharge to stakeholders in Australia so that depending on their risk aversion they can decide

to which future changes they want to adapt. Representation of spatial heterogeneous changes and their uncertainties is often done by showing maps with the mean of a multi-model ensemble as well as what fraction of the models agree on the sign of change (decrease or increase), such a representation of uncertainty is not very helpful for climate change adaptation, as a high agreement on the sign of change may be due to a range of model projections of, e.g., -20% to -30% or -5% to -50%. Model-based uncertainty of future changes of a variable for a certain spatial unit is often visualized by boxplots that show the

percentages of all ensemble members that exceed a certain change of the variable (e.g., Arias et al., 2021). Boxplots are challenging to comprehend due to the difficult interpretation and the non-unique definition of the "whiskers", one definition being the whisker length corresponding to 1.5 times the interquartile range; in addition, the handling of outliers is not fixed. To provide information about the range of possible future changes in meteorological changes in Germany, the Climate Service Center Germany (GERICS) uses bar charts with the whole range of changes projected by the different ensemble members,

showing, in addition, the median and the 20th (P20) and 80th (P80) percentiles, e.g., in their Climate-Fact-Sheets for Regions (available under https://www.climate-service-center.de/). Similarly, the letter-value plot shows several percentiles with bars, but with reduced bar width the more distant it is to the median. To show the distribution of values, violin plots can be used, which also show the minimum, median, and maximum values. Considering the numerous uncertainty visualization formats, a suitable format has to be identified for climate change adaptation processes.

Clear and precise communication on the uncertainty of future climate change is required to avoid biases and misunderstandings, and experts need to clarify the causes of the uncertainty and how uncertainty was determined  (Bles et al., 2019; Kloprogge et al., 2007). Experts should consider the so-called "usability gap", the gap between the information that knowledge producers (i.e., experts) perceive as *useful* and the information that knowledge users (i.e., stakeholders) consider *usable* in their daily work (Lemos et al., 2012). Only if knowledge users consider the information of, e.g., future hazards and

their inherent uncertainty as usable, they will include it in their decision-making process. "[..] [U]sability depends on three interconnected factors: users' perception of information fit; how new knowledge interplays with other kinds of knowledge that are currently used by users; and the level and quality of interaction between producers and users" (Lemos et al., 2012, p. 789). In general, multiple formats should be used to address a diverse audience, including words and numbers in graphs, along with narratives, images, and metaphors; when communicating with the general public, it is important to assume low numeracy

(Spiegelhalter et al., 2011). As the characteristics of the audience, such as a-priori beliefs and values, affect how uncertainty communication is received, communication should be adapted to it (Corner at al., 2018; van der Bless et al., 2019). Emphasizing uncertainty alone may discourage and create hesitancy among the audience, making it crucial to also explain areas of high scientific consensus (Corner et al., 2018). Scientists might be afraid that communicating uncertainty reduces trust, but the communication of "(epistemic) uncertainty does not always have a negative effect on people's affective states"

(van der Bles et al., 2019, p. 27). Finally, if experts communicate uncertainty issues well, stakeholders might gain an enhanced understanding and acceptance of model results as well as of adaptation measures (Parviainen et al., 2020).

The objective of this paper is to show how to quantify climate change hazards with their uncertainties for any region around the globe from publically available ISIMIP multi-model output, and how this information can be communicated in a participatory process as a starting point for identifying local climate change adaptation strategies. We utilize experiences from transdisciplinary research on freshwater-related adaptation to climate change in a biosphere reserve. The research project KlimaRhön aimed at developing, in a participatory process with local stakeholders, climate change adaptation strategies that enable society and freshwater ecosystems to sustainably use the changing water resources in the UNESCO biosphere reserve Rhön in Germany. In the project, the authors of this paper quantified and communicated future changes in total runoff and groundwater recharge and their shallow and deep uncertainties as derived from a freely globally available multi-model ensemble of GCMs driving global hydrological models (GHM) at the very beginning of the participatory process. The communication approach was evaluated by the participants of the participatory process as well as by the audiences of two presentations of the project results outside of the participatory process. In the two presentations, our approach for communicating uncertain climate change hazards was compared with a more common one.

The approaches for quantifying and for communicating the uncertainty of climate change-induced hydrological hazards are described in the next section. In Section 3, we present the evaluation of our communication formats. We discuss our results in Section 4 and finally draw conclusions in Section 5.

## 2 Quantifying and communicating the uncertainty of the climate change-induced hydrological hazards

Future changes should be quantified with their uncertainty and then a suitable visualization should be found with which the future changes with their uncertainty can be communicated in participatory climate change adaptation processes; an approach that we applied (Figure 1). At first, scientists or experts have to decide on what and how to produce climate change risk information (Figure 1, left box), and how to visualize the information (Figure 1, arrow between the boxes) before they communicate it to local stakeholders (Figure 1, right box). So, during the first step, the quantification, they need to decide what indicators of climate change hazard should be quantified, given the problem, the interest of the stakeholders, data availability and quality as well as technical and time constraints. In the second step, the scientists, experts or communicators have to decide on what, with which visualization format, and how to communicate given their audience, the aim and the generally severe time constraints in the participatory process. Thus, this approach partially follows the framework for uncertainty communication of "who communicates what, in what form, to whom and to what effect" (van der Bles et al., 2019, p. 3). In our study, we did not consider the option of not communicating our analysed results ourselves. In Section 2.1.2, we explain "what is communicated" (Figure 1, left box), and in Section 2.2, we elaborate on "in what form is communicated" (Figure 1, right box). It was evident "to whom" we communicated, i.e., that our communication targeted local stakeholders from our study area (Section 2.2.1), who possess diverse (experiential, educational and professional) backgrounds. Our objective ("to what effect is communicated") was to raise awareness about uncertainties and enable stakeholders to make more informed decisions in their

respective roles and engage better discussions during the subsequent workshops in the participatory process of the project KlimaRhön.

165   The terms and concepts used in the field of climate change risks and adaptation have different definitions in different contexts. To ensure mutual understanding and alignment, we recommend clarifying the definition of central ambiguous terms with stakeholders according to Table A1 in the Appendix.

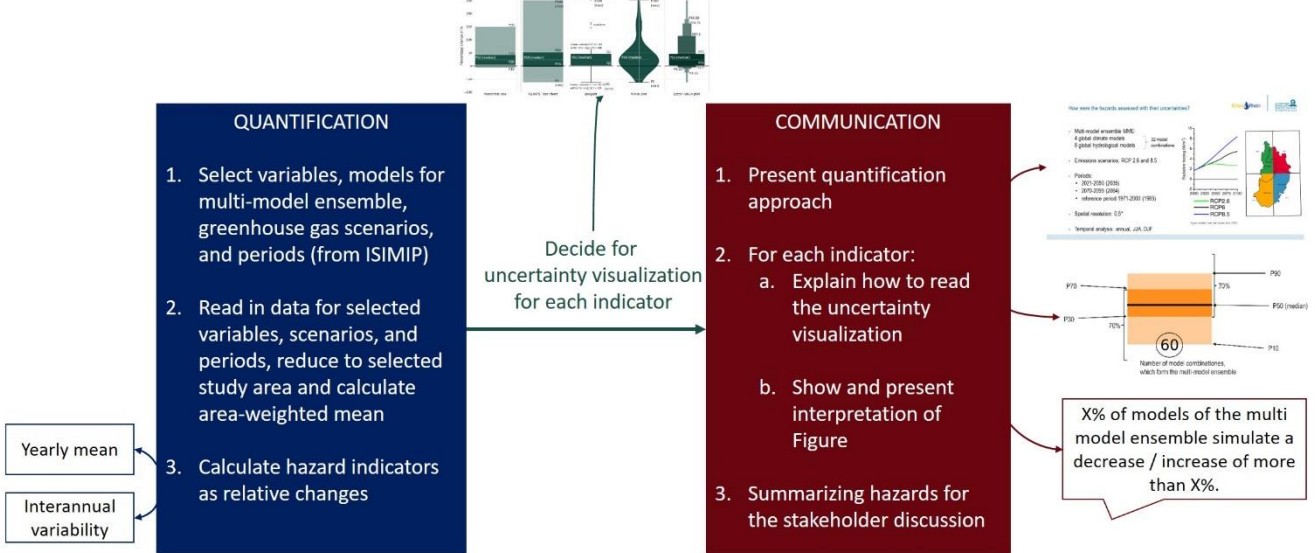

**Figure 1. Schematic of the presented approach of quantifying and communicating uncertain climate change hazards in participatory**
170 **climate change adaptation processes. ISIMIP: The Inter-Sectoral Impact Model Intercomparison Project (www.isimip.org/).**

## 2.1 Quantification of hydrological hazard indicators

### 2.1.1 Study area

The UNESCO biosphere reserve Rhön (BRR) is located in Central Germany at the border triangle of Hesse, Thuringia, and
175 Bavaria and covers an area of 2,433 km² (Biosphärenreservat Rhön, 2018; Figure 2). Numerous small and large water suppliers cover the local water demand mainly with spring water and groundwater and deliver water to surrounding regions (Schönthaler and Andrian-Werbung, 2008). The per-capita water demand is similar to the German average, and water resources are currently sufficient to fulfill the demand without overexploitation (Schönthaler and Andrian-Werbung, 2008). There are many springs, creeks and upper reaches of streams in the UNESCO biosphere reserve Rhön that are valuable aquatic ecosystems due to being
180 near-natural and due to their structural diversity; they are the habitat for many rare and protected species (Biosphärenreservat Rhön, 2018; Schönthaler and Andrian-Werbung, 2008). Until May 2018, 2,097 animal species were determined in 3,229 springs (Zaenker and Reiss, 2018), out of the estimated 10,000 springs in the BRR (Biosphärenreservat Rhön, 2018).

   Climate change leads to less snowfall in the BRR, and winter precipitation is potentially increasing, leading to more runoff in winter (Schönthaler and Andrian-Werbung, 2008). Summer precipitation is potentially decreasing and

185 evapotranspiration increasing, leading to lower surface water and groundwater levels and flows as well as to drying of springs in the summer, with negative effects on aquatic ecosystems and human water supply (Schönthaler and Andrian-Werbung, 2008). From 2018 to 2020, the BRR faced three extraordinarily dry summers. In interviews in October, November and December 2020 (Section 2.2.1), the stakeholders reported below-average streamflow, declining (ground)water levels, declining spring discharge, dried-out creeks and springs as well as ordinances to limit water use and difficulties in the public

water supply in some BRR regions (Rhön-Grabfeld).

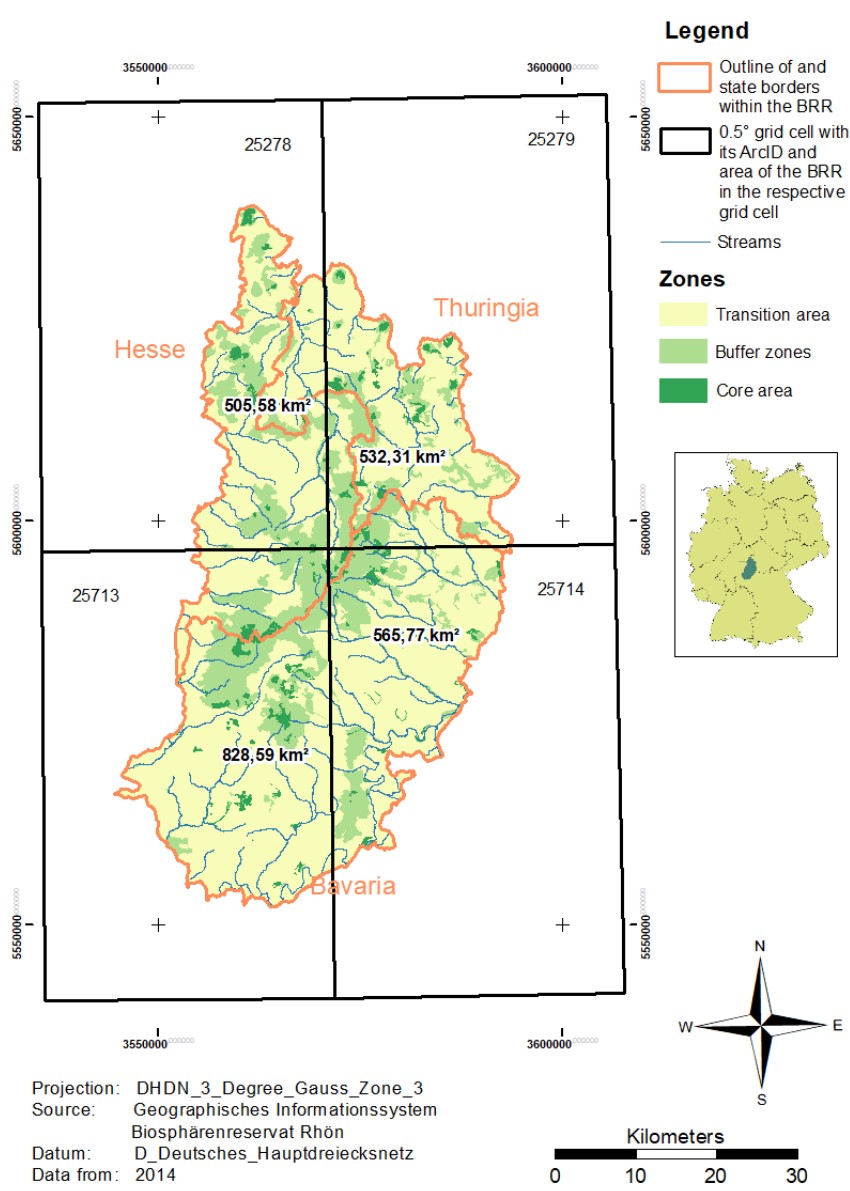

**Figure 2. Map of the study area, the UNESCO biosphere reserve Rhön (BRR), and its location in the 0.5° grid cells of the global hydrological models. The figure of the location of the BRR within Germany stems from a figure in Zaenker and Reiss (2018, p. 35).**

### 2.1.2 Processing of multi-model ensemble output

Changes in the hydrological variables total runoff and groundwater recharge were used to inform local stakeholders about potential future hydrological changes in the study area. The long-term average total runoff of a region corresponds to the renewable water resources in this region (Döll et al., 2015). It is comprised of two components, groundwater recharge and surface runoff (Ertl et al., 2019). Groundwater recharge is the component of total runoff that replenishes the groundwater, and its long-term average is the renewable groundwater resources (Ertl et al., 2019). Groundwater was analyzed in addition to total

runoff as water supply in the BRR mainly depends on groundwater.

To assess the potential impact of future climate change on total runoff and groundwater recharge in the BRR, we used the output of a GCM-GHM multi-model ensemble that consists of eight GHMs, each of which was driven by the bias-adjusted output of four GCMs, which resulted in an ensemble of 32 model combinations, i.e., ensemble members (for more detail see Appendix A3). These multi-model outputs are provided by ISIMIP (ISIMIP 2b, www.isimip.org; for more detail see Appendix

A4) and have a spatial resolution of 0.5° latitude by 0.5° longitude (55 km by 55 km at the equator) and varying temporal resolutions depending on the impact variable (ISIMIP, 2019). We used the output of the GCM-GHM multi-model ensemble for two reasons, 1) there exists no regional hydrological model that could be applied for climate change assessments and 2) the application of a single regional model might not decrease the uncertainty but lead to an underestimation of uncertainty as the uncertainty of hydrological models regarding the translation of climate change into hydrological change would be neglected

(Davie et al., 2013; Reinecke et al., 2021). We processed, analyzed, and communicated the output variables total runoff (daily resolution; variable name "qtot") to compute total water resources as well as groundwater recharge (monthly resolution; variable name "qr").

In the first step, we processed the data by reading in the NetCDF files for the hydrological variables total runoff and groundwater recharge, structuring the needed data, and saving it in an extra file. From the original ISIMIP NetCDF file, we

selected the data only of the grid cells overlying the BRR (Figure 2), two chosen greenhouse gas emissions scenarios, and for the reference period 1971-2000 and two future 30-year periods – the "near future around 2035" (2021-2050) and the "far future around 2084" (2070-2099), which were chosen with our project partners, the administrative offices of the BRR, to ensure comparability with their studies and a German climate projection ensemble project (Hübener et al., 2017; Appendix A5). Emissions scenarios represent the substantive epistemic uncertainty of future human activities affecting greenhouse gas

emissions (Döll and, Romero-Lankao, 2017; Table A1). We used the two representative concentration pathways RCP2.6 and RCP8.5 (for more detail see Appendix A6) to inform stakeholders about two possible courses of anthropogenic emissions.

In the next step, the monthly or daily data was aggregated to yearly averages of annual and seasonal flows. As for seasons, we decided on the summer months (June, July, August) and the winter months (December, January, February), because 1) in the BRR climate change tends to decrease summer runoff and increase winter runoff (Schönthaler and Andrian-

225 Werbung, 2008) and 2) there is a high public awareness for drying summers due to the recent summer droughts (Section 2.1.1).

To not overwhelm the audience, i.e., the stakeholders, with too much information, we decided to not analyze the fall and autumn months.

With these annual and seasonal yearly means, we first calculated area-weighted averages (Appendix A7) and then either a mean over each period or sorted the values of each period after magnitude for the analysis of interannual variability. The 30-year mean was calculated to give the stakeholders an overview of the future change tendencies. The 30-year mean values were calculated for the two future periods and the reference period with the annual averages as well as the seasonal averages of the summer and winter months. Climate change might lead to a higher interannual variability, which would mean that, e.g., the relative decrease of water resources during a dry year may be higher than the average decrease. To additionally show the stakeholders and make them aware of how interannual variability of total runoff and groundwater recharge may change due to climate change, all 30 yearly values of each 30-year period (the two future periods and the reference period) were sorted according to their magnitude resulting in exceedance probabilities (for more detail see Appendix A8).

In the next step, the future period mean values and sorted values of each future period were converted to percentage changes with the historical values (Appendix A9), because the analysis of relative changes is more robust than the analysis of absolute values, one reason being the low accuracy of GCMs and GHMs when simulating current climate conditions. In the last step, the values were partitioned into three different multi-model ensembles of projected changes. For the far future, the RCP2.6 multi-model ensemble with 32 model combinations and the RCP8.5 multi-model ensemble with 28 model combinations are used; one GHM did not provide simulations for RCP8.5 (Appendix A3). For the near future around 2035, we analyzed the results of RCP2.6 and RCP8.5 together as one multi-model ensemble of the total 60 ensemble runs. This is appropriate because climate change until 2035 does not depend much on the emissions scenario, and by combining the outputs the changes and their uncertainties are more robust.

## 2.2 Communication of climate change hazards

The goal is to communicate the computed hazard indicators, i.e., the potential future hydrological changes, in a way that they become usable information for stakeholders. This requires a good understanding of the uncertainties of the model results and the relevance of these uncertainties for climate change adaptation. Due to the diverse academic backgrounds of stakeholders in participatory processes, our objective was to effectively communicate the quantification and the potential hydrological hazard indicators in a way that would be also accessible to stakeholders with limited educational backgrounds.

We presented the quantification method and computed hazard indicators related to groundwater recharge and total runoff during a 30-minute plenary session to 31 stakeholders in the participatory process (Section 2.2.1). To compare this communication format, we presented the same hazard indicators and its analysis to two audiences interested in the results of the project KlimaRhön in 2023 (Section 2.2.1) in two alternative ways, the proposed way applied in the participatory process (Sections 2.2.2.1-2.2.2.3) and a more common way following the IPCC (Arias et al., 2021; Section 2.2.2.4).

### 2.2.1 Participatory process

The aim of the transdisciplinary project KlimaRhön was to develop freshwater-related strategies for adaptation to climate change, considering both the well-being of humans and of biota in springs and streams. An interdisciplinary team of two sociologists and us, two hydrologists, designed and conducted the participatory process. The stakeholders represent a wide range of sectors including agriculture, nature conservation, political decision-makers (mayors), administration, industry, water supply entities and non-governmental organizations. The key stakeholders are the three administrative offices of the BRR, one for each federal state.

The participatory process comprised several interviews, five workshops, and three focus groups. The first, second, and third workshops took place in February, May and October 2021 in the form of video conferences, while the fourth and fifth workshops took place in June and November 2022 in the BRR. Before the first workshop, interviews with 22 stakeholders were conducted where their problem perspective on climate change risks regarding freshwater was elicited. The aim of all workshops was that stakeholders jointly develop climate change adaptation strategies, learn about other perspectives and network.

In this paper, we describe and discuss the method for quantifying and communicating the potential hydrological changes to which the stakeholders may have to adapt to. The quantification of the potential hydrological changes and its communication method was setup disciplinarily by us, the hydrologists in the team, before the participatory process. The communication was only part of the first out of five workshops in the participatory process, which had the character of a kick-off meeting. 31 stakeholders participated to learn about potential future changes in renewable groundwater resources and total renewable water resources by a presentation and to identify a problem in the form of one or two specific adaptation field(s) in a World Café. The stakeholders came from diverse sectors (agriculture, fishery, nature conservation, political decision-makers (mayors), administration, industry, research, water supply entities, and non-governmental organizations). The stakeholders, except two, were not accustomed to working with climate change information and they were not familiar with assessing uncertainties using multi-model ensembles.

After the participatory process of the project KlimaRhön had been finished, we presented its results (including the quantified potential changes of water resources, workshop methods, and participatorily identified adaptation measures) in June 2023 to an expert group of 7 managers of inter-municipal alliances and municipal climate change managers (in the following referred to as in-person presentation). In July 2023, 67 persons attended an online presentation, in which we again presented the project results, that was open to everyone and free of charge and included mainly citizens, but also actors from administration as well as members of the administrative units of the BRR.

### 2.2.2 Communication of the quantification

In the first workshop of the participatory KlimaRhön process (Section 2.2.1), we first explained to the stakeholders why and how the uncertainty of future hydrological changes was quantified following Kloprogge et al. (2007) and why multi-model

ensembles represent the currently best estimate of future hydrological hazards. The slide used for the explanation can be seen

in Figure 3. Then, the potential changes of 30-year mean values was communicated by percentile boxes (Section 2.2.2.1) and the potential changes in interannual variability was communicated by continuous percentile boxes (Section 2.2.2.2). Finally, we summarized the hazards for the following stakeholder discussion (Section 2.2.2.3).

When presenting the project results after the participatory process, we also first explained why and how we quantified the uncertain future hydrological changes with the slide in Figure 3. Then, the potential changes of 30-year mean values was

295 communicated by percentile boxes (just like in the first workshop; Section 2.2.2.1) and, finally, an alternative communication of potential changes of 30-year mean values by tables was used to compare our communication approach by percentile boxes (Section 2.2.2.4).

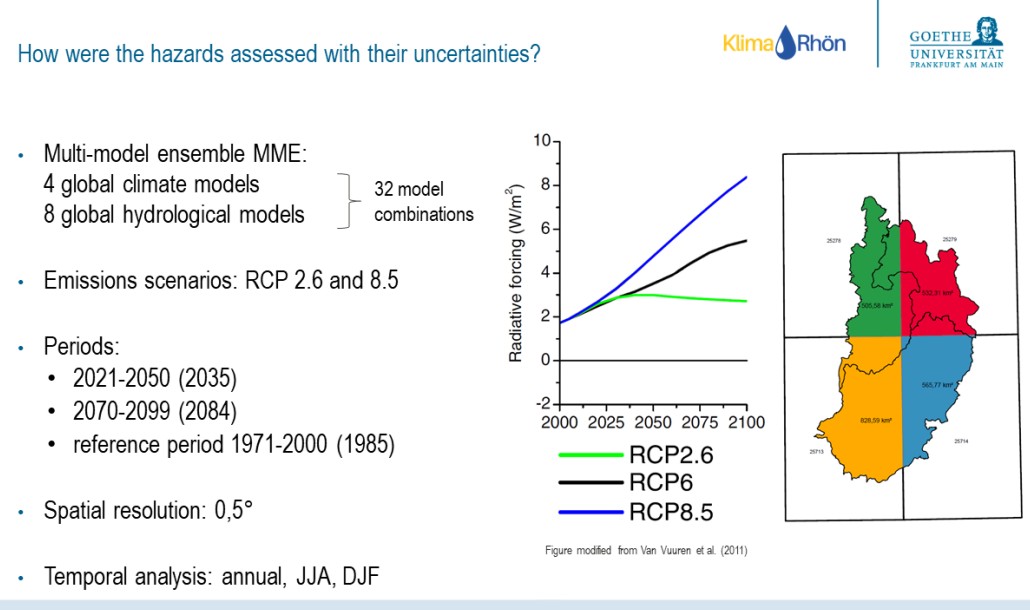

**Figure 3. With this slide (the original slide was in German), the quantification of the hydrological hazard indicators (Section 2.1.2)**
**was explained to the stakeholders in the first workshop. RCP: representative concentration pathway; JJA: June, July, August; DJF: December, January, February.**

### 2.2.2.1 Communication of potential changes of 30-year mean values by percentile boxes

To show potential changes of 30-year mean values, we designed graphics resembling those of GERICS (Section 1) that we refer to as "percentile boxes". Percentile boxes are similar to boxplots but easier to understand because of avoiding the boxplot

"whiskers" (Section 1). We selected five characteristic percentiles of the multi-model ensemble output to inform stakeholders about potential future changes in both total runoff and groundwater recharge. The characteristic values P10, P30, P50 (median), P70, and P90 represent the interpolated values of percent change of either total runoff or groundwater recharge that is not exceeded by 10%, 30%, 50%, 70%, and 90% of the 32 (or 28 or 60) values of the multi-model ensemble, respectively.

P10 and P90 form the outer margins of the percentile box, P30 and P70 the margins of the darker box inside the whole

percentile box, and P50 (the median) is displayed as a line within the percentile box (Figure 4). Thus, the projected changes of

80% of all ensemble members are represented by the percentile box; the number at the bottom of the percentile box shows the

number of model combinations in the multi-model ensemble. When informing about potential changes in 30-year mean values

in the first workshop, we first explained the percentile boxes with the characteristic values while showing Figure 4.

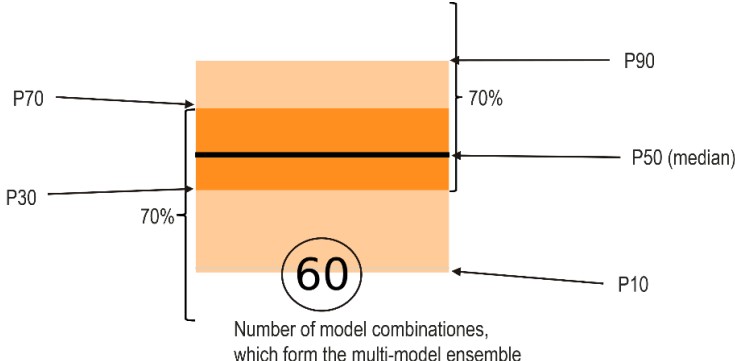

**Figure 4. Schematic for the explanation of the percentile box, the first diagram that was explained to the stakeholders in the first workshop. The characteristic values P10, P30, P50 (median), P70, and P90 represent the interpolated values of percent change of either total runoff or groundwater recharge that is not exceeded by 10%, 30%, 50%, 70%, and 90% of simulated values of the multi-model ensemble, respectively.**

After communicating how and why we quantified the hydrological hazard indicators with their uncertainty (Figure

3) and the percentile box (Figure 4), we presented the quantification results and clarified how the visualization can be

interpreted first with the results of the potential changes of 30-year mean values with their uncertainty showing Figure 5. As

an example, we explained the potential change of groundwater recharge in the period 2070-2099 (see the lower right diagram

in Figure 5) in this way:

"For the annual mean values, the dark grey box is never completely above or below 0% and the median lies

near 0%. This means that about 50% of the multi-model ensemble estimate a decrease and about 50% an increase. In

the summer and winter months, you can see a better tendency: The dark green box of the winter months is completely

above 0%. This means that at least 70% of the multi-model ensemble estimate an increase in the winter months. For

the summer months (under RCP 8.5), at least 70% of the multi-model ensemble estimate a decrease because the dark

orange box lies completely below the 0% line. This indicates a potential hazard, for, e.g., the drinking water supply

in summer months in the future. 10% of model combinations even project a decrease in summer groundwater recharge

by more than 60%."

We pointed out that none of the percentile boxes is completely above or below 0%. This means that in no examined case 90%

or more of the multi-model ensemble agree on the direction of change.

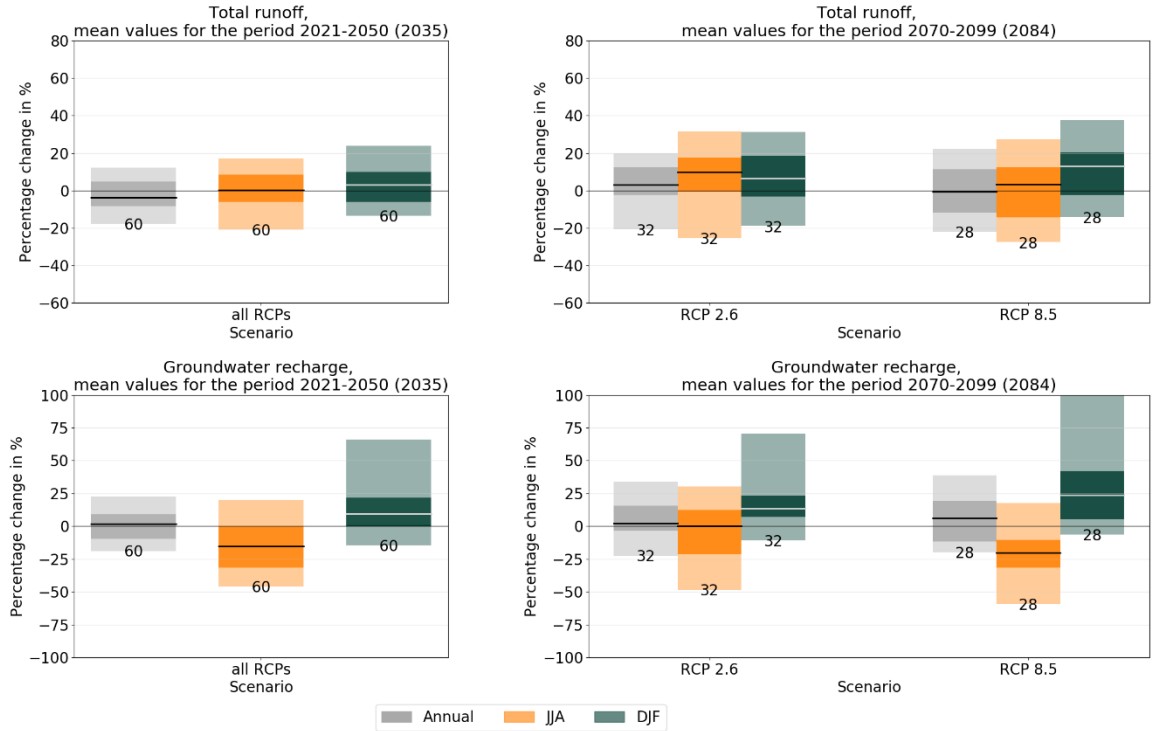

Figure 5. Percentile boxes showing the potential percentage changes of the mean total runoff and mean groundwater recharge of the multi-model ensemble in the periods around 2035 (2021-2050) and 2084 (2070-2099) relative to the reference period around 1985 (1971-2000) in the UNESCO biosphere reserve Rhön. The figure shows the annual and seasonal (summer months June, July, and August JJA as well as winter months December, January, and February DJF) means. For the period around 2084, the results are shown separately for the emissions scenarios RCP 2.6 and 8.5. In the period around 2035, the simulations of both emissions scenarios RCP 2.6 and 8.5 are shown together in one percentile box ("all RCPs"). The black numbers below the percentile boxes show the number of model combinations of the multi-model ensemble. For groundwater recharge in winter months under RCP 8.5 in the period around 2084, the percentile box was cut for better visualization, as its P90 lies at around 150% (Figure 10). This figure was shown to the stakeholders directly after Figure 4 and was explained like shown in lines 324-331. An interpretation of the results is given in Appendix A10.

## 2.2.2.2 Communication of potential changes in interannual variability

We also displayed the course of the five characteristic values over the sorted means (exceedance probabilities) of total runoff and groundwater recharge of each year in the two future periods. This shows the probability and uncertainty of changes in interannual variability, in particular how statistical dry years, with, e.g., very low groundwater recharge, may change in the future as compared to statistical wet years. We displayed the changes in groundwater recharge in wetter to dryer years in a "continuous percentile box". Two examples of a continuous percentile box were shown to the stakeholders in the first workshop and are shown in Figure 6, the x-axis indicates the exceedance probability of annual and summer groundwater recharge. For example, a value of 90% on the x-axis represents the annual / summer groundwater recharge in a rather dry year, a year with an annual / summer recharge that is exceeded in 90% of the 30 years of the reference and future period. The y-axis shows the percentage change of the annual / summer groundwater recharge between the reference period and the future period (2077-

355 2099). We told the stakeholders how they can relate the solid and dashed lines and the margins of a colored area in Figure 6 to the percentile boxes of Figure 5 shown before.

In the first workshop, after presenting the potential changes of 30-year mean values with their uncertainty (Figure 5), we explained the continuous percentile box with the characteristic values and then communicated the potential changes in interannual variability. We showed the changes of the interannual variability of groundwater recharge averaged over 1) all

360 months (Figure 6, left diagram) and 2) only the summer months (Figure 6, right diagram) in the years of the period 2070-2099 to the stakeholders, because we assumed the demand for adaptation in water management for groundwater recharge in the summer months higher than in the winter months after the stakeholder interviews. As an example, we explained the potential change of the mean summer groundwater recharge in the period 2070-2099 (Figure 6, right diagram) in this way:

"Here, in summer, let's focus on RCP 2.6 (green). The median drops to the right - on the right are drier

365 years! In wetter years (on the left), the median lies near 0%. So, about 50% of the multi-model ensemble estimate a decrease and about 50% an increase in wetter years. In drier years (on the right), you can see that even P70 - the upper dashed line - drops below 0%. This means that 70% of the multi-model ensemble estimate a decrease in summer months in drier years. Here, you see a difference in the consensus of the multi-model ensemble on the direction of potential changes between wetter and drier years. In drier years, the likelihood of an emerging hazard (less

groundwater recharge in summer months than historically) is higher than in wetter years."

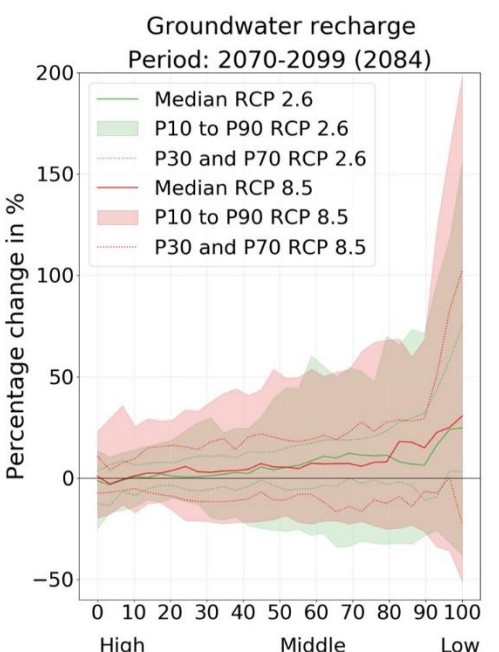 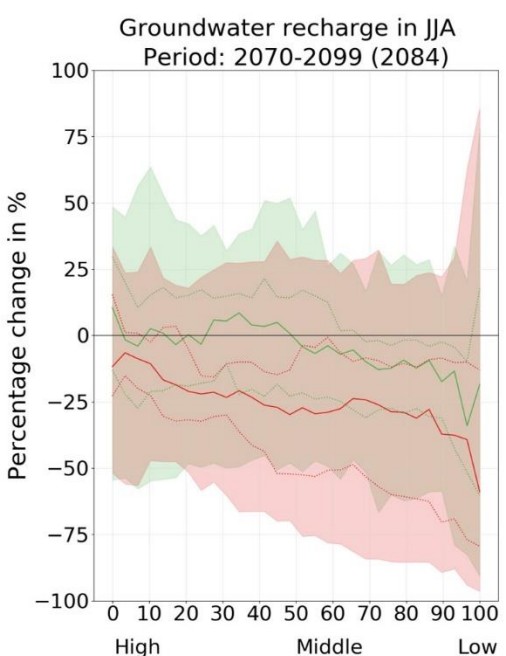

**Figure 6. The third diagram that was explained to the stakeholders in the first workshop, showing changes in the interannual variability of groundwater recharge. Percentage change of annual groundwater recharge (left diagram) and summer groundwater**

recharge in June, July, and August (JJA; right diagram) in the period around 2084 (2070-2099) relative to the reference period around 1985 (1971-2000) in the UNESCO biosphere reserve Rhön. The 30 yearly groundwater recharge values are sorted according to their magnitude and the relative changes were calculated between the values of the same rank in the future and reference period. This sorting corresponds to the exceedance probability, which is shown in percent. As an example, an exceedance probability of 90% in the left diagram refers to the year in which groundwater recharge is lower than in 90 % of all years, i.e., lower than in 27 out of 30 years. If the percentage change for this unusually dry year is -10%, this means that the model projects that the groundwater recharge that is exceeded only in 3 out of 30 years in the future is 10% lower than in the respective year of the reference period. This figure was shown to the stakeholders directly after Figure 5 and was explained like shown in lines 364-370. An interpretation of the results is given in Appendix A11.

### 2.2.2.3 Summarizing hazards for the stakeholder discussion

After the presentation of the potential changes of the 30-year mean values and the interannual variability with their uncertainty in the plenary, we organized a World Café. On one of the tables, all the stakeholders discussed for about 15 minutes the following question: "What future climate change-driven changes of groundwater recharge and runoff, which can only be estimated with uncertainty, do we want to prepare for?" This was to determine the risk aversion or affinity of the stakeholders. To support the discussion, summarizing the multi-model results shown in the plenary, we showed the following information at the World Café table:

- Potential change in mean annual total runoff: - 20% to + 20%, median near 0%

- Potential change in mean annual groundwater recharge: - 25% to + 25%, median near 0%

- Potential change in mean groundwater recharge in summer months (June, July, August): - 70% to + 25%, median at approximately - 20%, groundwater recharge especially declines in dry years

The ranges correspond approximately to the P10 and P90 values. We did not present stakeholders with the lowest value, i.e., the strongest simulated decrease of the multi-model ensemble. The worst-case scenario, which we presented, included percentage reductions that are exceeded by only 10% of the multi-model ensemble, representing the smallest values within the three ranges mentioned above.

### 2.2.2.4 Alternative communication of potential changes of 30-year mean values by tables

The tables shown in Figure 7 were presented to the audiences of the two presentations (outside of the participatory process) to compare our visualization format (Section 2.2.2.1) with this alternative communication format. The tables represent the same data on future groundwater recharge and runoff changes that is shown by the percentile boxes (Figure 5) in an alternative, simpler and more common format, in the form of a table with numerical values, the mean change of the ensemble members, and verbal expressions describing the agreement of the ensemble members on the sign of change. If more than 80% of models agreed on the sign of change, we assigned high agreement. If 60-80% of models agreed on the sign of change, we assigned medium agreement, and if less than 60% of models agreed on the sign of change, we assigned low agreement.

Potential changes of total runoff and groundwater recharge 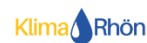 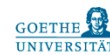

Potential changes of the 30-year mean of **total runoff** compared to 1971-2000:
Mean of the multi-model ensembles & agreement on the sign of change

| | 2021-2050 | 2070-2099 | |
|---|---|---|---|
| Emissions scenario | all RCPs | RCP 2.6 | RCP 8.5 |
| Annual | -3%    (low) | +2%    (low) | +1%    (low) |
| Summer months | -0%    (low) | +6% (medium) | +1% (medium) |
| Winter months | +3%    (low) | +9% (medium) | +11% (medium) |

Potential changes of the 30-year mean of **groundwater recharge** compared to 1971-2000:
Mean of the multi-model ensembles & agreement on the sign of change

| | 2021-2050 | 2070-2099 | |
|---|---|---|---|
| Emissions scenario | all RCPs | RCP 2.6 | RCP 8.5 |
| Annual | -1%    (low) | +5%    (low) | +8%    (low) |
| Summer months | -16% (medium) | -7%    (low) | -19% (medium) |
| Winter months | +14% (medium) | +19% (medium) | +40% (medium) |

Agreement of the models on the sign of change
low: <60% of models of the multi-model ensemble
medium: 60-80% of models of the multi-model ensemble
high: >80% of models of the multi-model ensemble

**Figure 7. With this slide (the original slide was in German), the change in mean total runoff and mean groundwater recharge in the periods 2021-2050 (2035) and 2070-2099 (2084) as compared to 1971-2000 for the whole year (annual), only the summer months and only the winter months was shown in an alternative way as compared to the percentile boxes (Figure 5). For the period 2070-2099, the results are shown separately for the emissions scenarios RCP 2.6 and 8.5. In the period 2021-2050, the simulations of both emissions scenarios RCP 2.6 and 8.5 are shown together in one column ("all RCPs"). Agreement within the multi-model ensemble on the sign of change is provided in parentheses: in the case of low model agreement <60% of models agree on the sign of change; in the case of medium model agreement 60%-80% of models agree on the sign of change; in the case of high model agreement >80% of models agree on the sign of change. This slide was not shown to the stakeholders in the first workshop of the participatory process.**

## 3 Results

### 3.1 Interpretation of communicated hazard indicators by the stakeholders

In the discussion about what degree of changes the BRR should aim to adapt to (Section 2.2.2.3), all except one advocated for adapting to the worst-case (the strongest decreases presented were P10), driven by the precautionary principle and the recognition of the time required for practice adjustments. These reductions include groundwater recharge decreases of over 70% in summer and annual groundwater recharge decreases of 25%. However, concerns were raised regarding the potential cost and frustration associated with preparing for worst-case scenarios that might not materialize. Another stakeholder asked herself how important it is to decide to adapt to a specific (i.e., deterministic) potential change, arguing that a wrong certainty was conveyed if they decided to adapt to a specific potential change. On the other side, it was discussed that for some (technical) adaptation measures, a specified quantity of potential change is needed. Moreover, they highlighted the limitations of mean annual total runoff as an informative indicator and instead emphasized the importance of focusing on extreme events. In summary, the need for anticipatory, flexible and robust measures in response to uncertainties was stressed, with stakeholders

being very risk-averse. The presentation of potential future groundwater recharge changes in summer months and unusually dry years, in particular summers, in the study area and all of Germany before and during the participatory process led to a focus on adapting to potential water scarcity in summer months.

**3.2 Evaluation of the communication format by the stakeholders of the KlimaRhön participatory process**

At the end of the first workshop, the stakeholders anonymously evaluated the applied methods for embracing uncertainty (the evaluation data can be found in Müller and Czymai, 2022). The stakeholders were asked to rate their agreement on the statement "Through the scientific input, I gained a better understanding of the potential changes to water resources due to climate change in the biosphere reserve Rhön". The scientific input consisted of the explanation and discussion of simulated

mean and interannual changes of total runoff and groundwater recharge as presented in Sections 2.2.2.1, 2.2.2.2 and 2.2.2.3. For more than 60% of the 26 stakeholders, the scientific input enhanced the understanding of potential changes in water resources while 12% of stakeholders rather disagreed with the statement, and no stakeholder strongly disagreed (Figure 8, upper left bar). Even more than 75% of the stakeholders strongly agreed or rather agreed with the statement "Through the scientific input, I gained a better understanding of the uncertainties of the estimated changes in the biosphere reserve Rhön" (Figure 8, lower left bar).

(Figure 8, lower left bar). Only one stakeholder rather disagreed with the statement and none strongly disagreed.

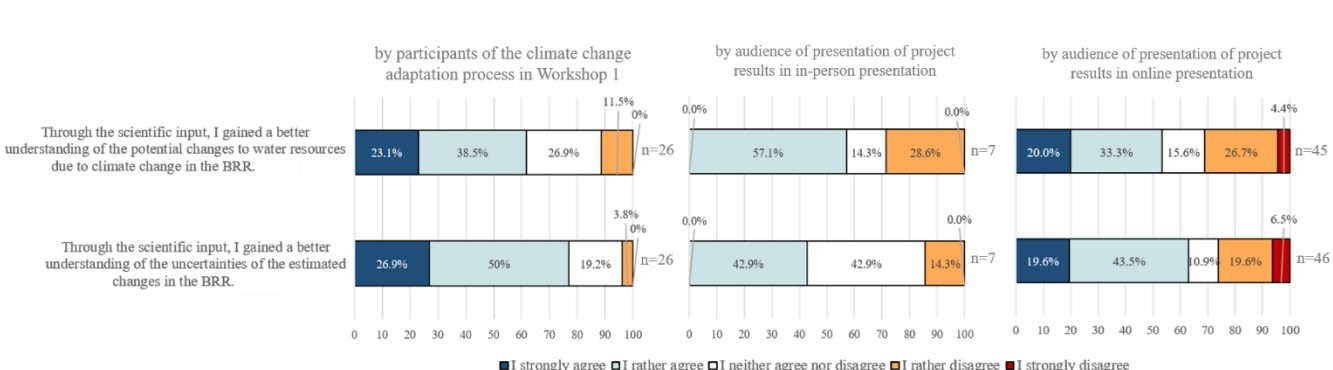

**Figure 8. Results of the evaluation of the scientific input in the first workshop of the participatory process (left bars), and in the in-person presentation (middle bars) and online presention (right bars) outside of the participatory process. The stakeholders and the audiences of the project presentations rated their agreement on the statements "Through the scientific input / the results (figures) just presented, I gained a better understanding of the potential changes to water resources due to climate change in the biosphere reserve Rhön" (upper bars) and "Through the scientific input, I gained a better understanding of the uncertainties of the estimated changes in the biosphere reserve Rhön" (lower bars). The bars show the results of the answers of 26 stakeholders (left bars), 7 participants (middle bars), and 45 and 46 participants (right bars). BRR: UNESCO biosphere reserve Rhön.**

**3.3 Comparison of our communication format with a more common communication format by the audiences of two**
**presentations of the project results**

To compare the communication format of potential hydrological changes, more precisely of changes of 30-year mean values, by percentile boxes (Figure 5) with a more common communication format (Figure 7), we used the opportunity of two

presentations, in which we presented the outcomes of the project KlimaRhön at the end of the project (June and July 2023; Section 2.2.1). In each of the two presentations, we first presented the potential future changes in mean groundwater recharge and runoff in the same way as was done in the first workshop of the participatory climate change adaptation process, described in Section 2.2.2.1. Directly afterwards, we asked the audience to anonymously rate their agreement on the same two statements rated by the KlimaRhön stakeholders in the first workshop of the participatory process. For the first statement, "Through the results (figures) just presented, I gained a better understanding of the potential changes to water resources due to climate change in the biosphere reserve Rhön", only 57% of the audience of the in-person presentation rather agreed, while 28% rather disagreed (no one strongly agreed or strongly disagreed; Figure 8, upper middle bar). 53% of the audience of the online presentation strongly or rather agreed to the first statement, while 31% strongly or rather disagreed (Figure 8, upper right bar). For the second statement, "Through the results (figures) just presented, I gained a better understanding of the uncertainties of the estimated changes in the biosphere reserve Rhön", 43% of the audience of the in-person presentation rather agreed, while 14% rather disagreed (Figure 8, lower middle bar). 63% of the audience of the online presentation strongly or rather agreed to the statement (Figure 8, lower right bar). Like in the first workshop of the participatory process, more participants of the online presentation agreed to have gained a better understanding of the uncertainties than to have gained a better understanding of the potential changes (Figure 8, right bars). The evaluations of our communication format by percentile boxes differ between the groups (stakeholders in the workshop, audience in the in-person presentation and audience in the online presentation) because the evaluation in the workshop of the participatory process was conducted after the stakeholders were also shown the interannual variability changes (Figure 6), summarized multi-model results (Section 2.2.2.3) and then had time to discuss the potential hydrological changes in the World Café (Section 3.1), while the evaluations in the two presentations were conducted directly after communicating the 30-year mean values with its uncertainty. In addition, the participant composition of the groups was very different.

Directly after this evaluation (Figure 8, middle and right bars), the tables shown in Figure 7 were presented to the audiences of the two presentations. Then, we asked the audiences to anonymously indicate which of the two visualizations was better suited to improve their understanding of the potential changes and their uncertainties, the percentile box figures (Figure 5) or the tables (Figure 7). For the first question, "Which of the two visualizations (figure or tables) was better suited to improve your understanding of the potential changes to water resources due to climate change in the BRR?", and the second question, "Which of the two visualizations (figure or tables) was better suited to improve your understanding of the uncertainties of the estimated changes?", 57% of the seven participants of the in-person presentation preferred the tables and only 28% the percentile box (Figure 9, left bars). Similarly, 56% of 41 participants of the online presentation preferred the table and 29% preferred the percentile boxes for improving the understanding of the potential changes (Figure 9, upper right bar). However, for the second question on understanding the uncertainties, only 37% of the audience of the online presentation preferred the tables and 39% preferred the percentile boxes (Figure 9, lower right bar).

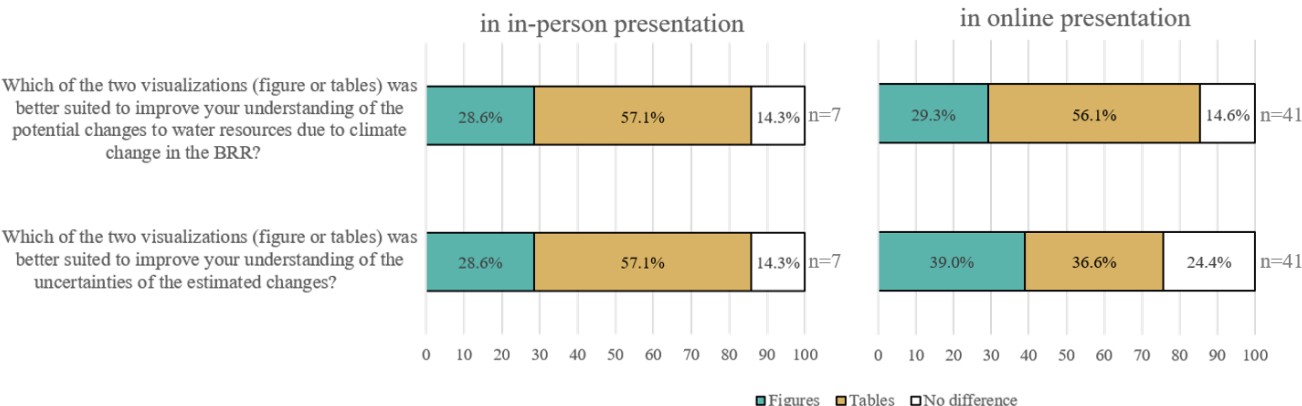

**Figure 9. Comparison of visualization formats in the in-person presentation (left bars) and in the online presentation (right bars) outside of the participatory process. The bars show the results of the answers of 7, respectively 41 participants. BRR: UNESCO biosphere reserve Rhön.**

## 4 Discussion

### 4.1 Why and how should the uncertainty of hydrological changes due to climate change be quantified with a multi-model ensemble of global models?

Uncertainty of future climate change hazards is expected to be high due to the complexity of the Earth system and human decision-making, highlighting the need to embrace uncertainty and to work on reducing the uncertainty of projected climate change and the related hazards (Mearns, 2010). Both global climate models and global hydrological models have a low spatial resolution and do not consider the high spatial heterogeneity that can be important for local scale adaption to climate change. Higher-resolution modelling efforts, if restricted to one local hydrological model, might be better suited for simulating the historic development in a study region, but are expected to lead to an underestimation of uncertainty regarding future changes. One reason is that the (again uncertain) impact of adapting vegetation on water resources (Reinecke et al., 2021) is mostly not taken into account by hydrological models. Therefore, we suggest using, under most circumstances, the available multi-model ensemble output of the ISIMIP initiative to quantify potential future changes of physical variables and their uncertainty, in a similar fashion as we did in the presented case study and not use a single local or regional impact model. A multi-model ensemble including regional models (combinations of regional climate and regional impact models or of global climate and regional impact models) would be favoured over a multi-model ensemble of only global models. While the coarse resolution of the global multi-model ensembles of the ISIMIP initiative certainly leads to increased and non-quantifiable uncertainty, a major advantage is its usability for participatory climate change adaptation processes all around the globe. We found that even with the coarse resolution of the results, they helped the stakeholders to understand future water-related hazards of climate

change and their uncertainty in the study regions, and based on this to focus their search for adaptation measures. However, it needs an expert with basic technical knowledge in any programming language to quantify the potential changes.

We considered the three complementary paradigms for modelling the future by Maier et al. (2016): Anticipation of the future was done by the GCMs and GHMs used, (approximate) quantification of future uncertainty was achieved by using multi-model ensembles, and the exploration of multiple plausible futures was done by using two emissions scenarios.

The monthly time series of a large number of hydrological variables that are provided in ISIMIP should be used to compute hazard indicators that are most relevant for the climate change risks of interest. For example, while groundwater recharge (renewable groundwater resources) is a fraction of total runoff (renewable water resources), change in mean groundwater recharge differs appreciably from projections of total runoff, in particular in the summer months, thus it is better to analyse groundwater recharge projections if groundwater is important for human water supply. Changes in variability and extremes should also be analysed specifically as it is generally assumed that climate change changes variability and thus, e.g., drought and floods. Additional hazard indicators should be analysed after major risks have been identified together with stakeholders based on the vulnerabilities of the system of risk. This may also include different spatial and temporal aggregations of the multi-model ensemble output, e.g., regarding seasons.

Results of multi-model ensembles may underestimate the uncertainty, which was shown with an initial condition large ensemble by Mankin et al. (2020). This initial condition large ensemble consisted of seven climate models which were run with different initial conditions and simulated a larger uncertainty range than CMIP5, an ensemble of 40 climate models. Such initial condition large ensembles, however, are not well feasible especially for climate change impact studies because of the large computational resources to run the model combinations (as in the example of Mankin et al., 2020: 286 climate model runs, which would then be multiplied with the number of impact models). Moreover, the initial condition large ensemble would also need to embrace parameter uncertainty, which would increase the model runs.

## 4.2 How can the uncertainty of hydrological changes due to climate change be best communicated?

Depending on the communication objective and the degree of intended precision, uncertainty can be communicated and visualized among others with percentile boxes, the bar charts used by GERICS (which are percentile boxes, but with different percentiles used), boxplots, violin plots, or letter-value plots (Figure 10). For a low degree of required uncertainty precision, only the range of uncertainty with the minimum and maximum values could be communicated. Higher precision can be achieved by subdividing the total range into percentiles (GERICS bar charts, percentile boxes, boxplots) that show the change values that are not exceeded by a certain percentage of the ensemble members. Uncertainty is visualized most precisely by violin plots, which display the distribution of values within the whole range by the width of the box; smoothing of the shape, however, leads to a only approximate representation of the distribution. The higher the communicated uncertainty precision, the more information is transported, which again might be overwhelming for the end user.

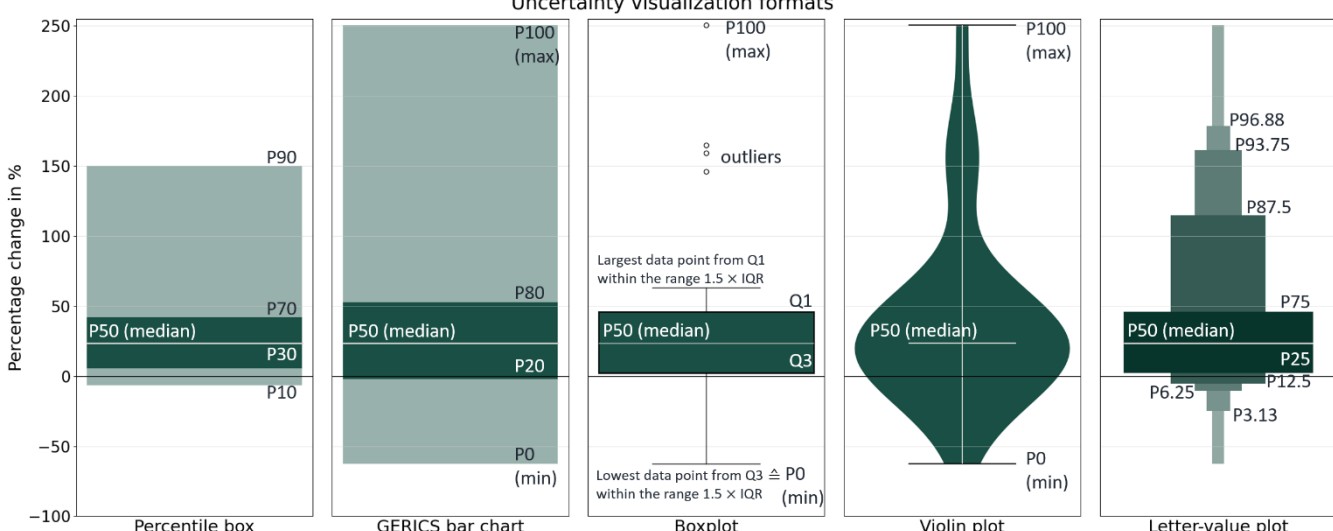

Figure 10: The uncertainty visualization formats percentile box, GERICS bar charts, boxplot, violin plot, and letter-value plot as an example for the potential percentage change of groundwater recharge in the winter months in the far future around 2084 under the emissions scenario RCP 8.5 simulated by 28 multi-model ensemble members. The script and an example data set for this comparative figure of uncertainty visualization formats for the same data is freely available (Müller, 2023). Q1: first quartile (≙ P25); Q3: third quartile (≙ P75);  IQR: interquartile range, i.e., range between Q1 and Q3.

We came up with our approach of the percentile boxes because we did not want to communicate minimum and maximum values 1) to not give too much room to possible outliers and 2) to not give the impression that minimum and maximum values could not be exceeded in reality. We wanted to have several percentiles to transparently visualize uncertainty to stakeholders, and to enable communication of more thresholds than other visualization formats. In this way, the communicator has the possibility to say how many percent of models simulated a stronger or weaker change than a certain

change value, which supports communication that does not include a specific prediction. Another advantage is that the percentiles can be chosen individually depending on the risk aversion and the problem (Table 1). Stakeholders would like to know about (almost) worst-case future changes, but maybe not the most extreme results and thus outliers of the ensemble. For our percentile boxes, we arbitrarily defined the upper and lower 10% of the multi-model results as outliers and therefore did not display them.

Table 1 shows the advantages and disadvantages of visualization formats (inlcuding the percentile box) that we identified for the communication of uncertainty to stakeholders. To improve the letter-value plot to communicate uncertainty in participatory processes, less and only selected percentiles should be shown, however, it is still misleading concerning the distribution of values (Table 1). When displaying minimum to maximum values and if the focus of the communication is on the distribution of values, we would recommend using violin plots to prevent outlier values from being misinterpreted as being

as frequently simulated as values within the P10 to P90 range. As the violin plot only indicates three percentiles, the median, minimum and maximum values, with a line, it may be more easily induce the viewer to interpret the median as best estimate

and disregard the uncertainty in the consequent adaptation decisions. Therefore, violin plots are less suitable for participatory processes than percentile boxes. But suitability can be improved when the violin plot is supplemented with more percentiles with lines. The percentile box, GERICS bar chart, boxplot, and letter-value plot could be supplemented with points displaying the change values to show the distribution of values.

**Table 1. Advantages and disadvantages of some visualization methods to communicate uncertain changes to stakeholders. All represent the distribution of changes that are simulated by the different members of the multi-model ensemble and indicate the change values that are not exceeded by a certain percentage (percentile) of the ensemble members.**

| Uncertainty visualization | Advantages | Disadvantages |
|---|---|---|
| Boxplot | Shows three percentiles (P25, P50, P75) and possibly outliers and minimum (P0) and maximum (P100) values; common visualization. | Potentially reduced readability if it shows minimum and maximum; non-unique definition of and difficult to interpret "whiskers"; does not show percentiles suitable for strongly risk-averse stakeholders (P10 and P90); does not show the distribution of simulated change values precisely. |
| GERICS bar chart | Shows five percentiles (P0, P20, P50, P80, and P100). | Potentially reduced readability as it shows minimum and maximum; does not show the distribution of simulated change values precisely. |
| Violin plot | Precisely visualizes uncertainty by displaying the distribution of simulated change values within the whole range by the width of the box. | Potentially reduced readability as it shows minimum and maximum; only shows the percentiles minimum, maximum, and median (does not show many percentiles); smoothing of the shape leads to only an approximate representation of the distribution; the high information content due to the high uncertainty precision might be overwhelming and difficult to read for the end user. |
| Letter-value plot | Shows many percentiles and uses the width of the boxes to guide the eye. | Potentially reduced readability as it shows minimum and maximum; area of the boxes does not correspond to the number of simulated change values contained in the boxes; pseudo- (i.e., misleading) visualization of the distribution of simulated change values; depending on the visualization, it has too many percentiles. |
| Percentile box | Shows five percentiles; percentiles can be selected depending on the problem and the risk aversion; minimum and maximum can be avoided, which increases the readability of the more central values and prevents the misinterpretation that the minimum and maximum values of the ensemble are the most extreme values possible. | Does not show the distribution of simulated change values precisely. |

Visualization of uncertainty by percentile boxes enables stakeholders to read the change values they wish to adapt to, depending on the problem (resource scarcity vs. resource excess) and their risk aversion (low vs. high). For example, a

stakeholder with a high risk aversion towards reduced water availability can choose P10 of 30-year mean change in summer months (from the percentile boxes of Figure 5) or even P10 in summer months for statistical dry years with an exceedance probability of 90% (continuous percentile box of Figure 6) under the emissions scenario RCP8.5. With our proposed types of visualization, percentile box and continuous percentile box, stakeholders should be empowered to recognize the uncertainty and probability of potential changes on the one hand for a future period on average and on the other hand for years with different water availability in a future period. However, the communication, i.e., the explanation of the approach and the interpretation, takes a lot of time in a workshop and asks the stakeholders for a long concentration span. To further increase the information content of percentile boxes, the numerical value of the change at each of the five percentiles could be written on or next to the respective line, which, however, would require larger percentile boxes. The visualization of changes in interannual variability by the continuous percentile box remains rather difficult to understand and requires, for most stakeholders, a longer exposure than was possible during the workshop. In addition, for Figure 6 we would propose to change the colors to such that are better visible for persons with color vision deficiency and thicken the lines of the inner percentiles for better visibility.

To avoid the wrong interpretation that the percentiles reflect true probabilities of occurrence (while they are only a very rough approximation of probabilities, Döll et al., 2015), we did not use the wording "with a probability of", also following Jack et al. (2020). The hazard uncertainty was suitably communicated in the participatory KlimaRhön process by always referring to the fraction of the ensemble members that projects changes of less or more than a certain value. By using the fraction of the ensemble members instead of terms like "the majority" or "most of the models", it may have prevented subjective misinterpretation of the actual fraction. For instance, some individuals might perceive 60% as "the majority", whereas some scientists consider 90 to 100% as representing a consensus (van der Bles et al., 2019). In addition, the communication of potential changes in groundwater recharge and total runoff might be more effective when they are related qualitatively to possible impacts and the reality of life of the stakeholders (Corner et al., 2018). We suggest giving examples of the potential impacts connected to the indicated hazards, e.g., the impacts of reduced summer groundwater recharge on drinking water supply or on the discharge of springs with related ecological impacts, and show illustrative images (e.g., from https://climatevisuals.org/; Corner et al., 2018).

The stakeholders and the audience of the online presentation rated their enhanced understanding of the potential changes somewhat lower than their enhanced understanding of uncertainty (Figure 8). This might be due to the large uncertainties and the fact that the multi-model ensemble of potential changes never fully agreed on the direction of change. Comparing our communication of potential changes of 30-year mean values by percentile boxes (Section 2.2.2.1) with a more common communication of the means of the multi-model ensembles in tables (Section 2.2.2.4), the audiences in the in-person and the online presentation preferred the more common visualization (tables) for an enhanced understanding of the potential hydrological changes. However, for improving their understanding of the uncertainties of these potential changes, an additional 20% found the figures to be more suitable or equally suitable as the tables compared to improving their understanding of the potential changes themselves (Figure 9). Of those 20%, half prefer our communication format with the figure, i.e., they find

that the more common communication with the tables helps to understand the potential hydrological changes but did not adequately account for uncertainties. The other half is undecided on whether the figure or the tables are more suitable for uncertainty communication. Possibly, our communication with the percentile boxes did not fit with the data normally used by the participants (lack of interplay; Lemos et al., 2012) or their (work) experience (tacit knowledge; Höllermann and Evers, 2019). We suspect that participants might have preferred the percentile boxes if 1) they would have had more time to interpret them and 2) they would have needed to use the hazard information for defining adaptation measures. We believe that our approach for communicating uncertainties in the stakeholder workshop of the participatory process was suitable and at an appropriate level of complexity, also because more than 75% of the stakeholders and 63,1% of the audience in the online presentation (Figure 8) agreed to have gained an enhanced understanding of the uncertainties. However, it must also be mentioned that this positive evaluation regarding their better understanding of uncertainties might have also been caused by a bias. They might have either wanted to give "socially wanted answers or [they] might [have wanted] to give themselves the idea that their time was well spent" (Kok and van Vliet, 2011, p. 102).

To tackle uncertainty, practitioners already have different routines but usually do not analyze model ensembles and their uncertainty themselves (Höllermann and Evers, 2019). By showing the stakeholders the uncertainty of the models through multi-model ensembles and showing them the approach of our quantification, we address one of the uncertainty routines of stakeholders called "transparency", in which the stakeholder considers the limits of knowledge (Höllermann and Evers, 2019). Moreover, decision-makers have to perceive the information as accurate, credible, salient, timely and useful for the decision-making need and they should not perceive it as risky to use the information (Lemos et al., 2012). Cash et al. (2002) found that the critical determinants of information for decision-making are credibility, salience and legitimacy, of which all must be fulfilled, but that decision-makers (or in general audiences) differently perceive and value these attributes. We think that we made the simulation results more credible and salient and as low risk as possible by showing uncertainty with the results of a multi-model ensemble in the percentile boxes and explaining them.

## 4.3 Evaluations in participatory processes

A research gap exists in how to best evaluate participatory methods. The evaluation conducted in this study is only a weak form of evaluation, as is often the case in participatory processes. In participatory processes, it is not practicable or even possible to form two groups that are submitted to "alternative treatments" such as in, e.g., clinical studies, because two parallel time-consuming participatory processes would have to be organised. We did not want to disturb the participatory process and burden the stakeholders with scientific investigations; hence, we used the opportunity of the two presentations of project results after the end of the participatory process for a comparative evaluation. This evaluation was not done by presenting each of the two communication formats to a different group, as we could not expect that the audiences would be comparable; instead, the two formats were presented and evaluated one after the other. However, the evaluation is only a weak evaluation in the sense that it did not evaluate the formats in the context of a participatory process on climate change adaptation but within a presentation of project results. The information needs of stakeholders that need to identify adaptation measures in response to

the presented climate change hazards are different from the information needs of audiences of a presentation of project results.

In the workshop, the hydrological changes due to climate change with their uncertainties (Figures 5 and 6) were communicated to the stakeholders to support their development of adaptation measures, which was evident to the stakeholders when listening to the scientific input. The audience of the presentations did not have to develop adaptation measures, so a less detailed and thus simpler communication format (Figure 7) than the percentile box (Figure 5) was sufficient, but would not be sufficient as a basis for identifying adaptation measures. Moreover, the time that the stakeholders dealt with the information at the workshop

was much longer. The evaluation in the workshop was conducted only after the stakeholders were also shown the interannual variability (Figure 6), the most important hydrological change ranges (Section 2.2.2.3) and then had time to discuss the potential hydrological changes in the World Café (Section 3.1), while the evaluations in the two presentations were conducted directly after communicating the 30-year mean values with its uncertainty with the two communication methods. Therefore, the comparative evaluation of the two communication formats by the audiences of the presentations is not relevant for the

communication format in participatory climate change adaptation processes.

### 4.4 Using the uncertain information about future climate change hazards for the development of adaptation measures

The stakeholders in the first KlimaRhön workshop expressed a preference for following the precautionary principle and wanted to adapt to a worst-case scenario regarding water scarcity in the summer months (a strong decrease in groundwater recharge) rather than to an increase in groundwater recharge, even though this was within the simulated ensemble range and the median was close to zero. At this early point, they did not see the need to agree on adapting to a specific future change of groundwater

recharge (Section 3.1) because adaptation measures were only generally discussed. The specific results of the multi-model ensemble were not used quantitatively (only qualitatively) in the discussion of adaptation measures because no technical measures (e.g., well drilling and networking of pipelines) that must be based on a specific decrease of groundwater recharge were planned, and no monetary cost-benefit analyses were performed in the participatory process. With our percentile boxes,

decision-makers can use the provided range of potential changes in, e.g., exploratory modelling to stress test the system with different plausible futures and possible decisions, which would not be possible with the results in the more common communication format (Figure 7).

      Communicating and embracing uncertainty is important "to help policy makers and practitioners make the best possible decisions, which cannot be based on the available evidence alone. […] [T]aking unknowns into account aims to allow

more realistic assessment of the adequacy of decisions, as well as better preparation for things that can go wrong" (Bammer, 2013, p. 64). Due to highlighting the uncertainty of future changes, we hope that the stakeholders will more carefully embrace uncertainty in their decision-making in the future. Next to the uncertainty of climate change hazards (comprising the climate and hydrological model uncertainty and the uncertainty of greenhouse gas concentrations), the uncertainty of suitable adaptation measures, i.e., uncertain transformation knowledge (Becker, 2002), persists, as multiple plausible futures could

unfold. This uncertainty should be embraced, e.g., with participatory scenario development (Carlsson-Kanyama et al., 2008, Maier et al., 2016; Döll and Romero-Lankao, 2017; Voinov et al., 2018), without overwhelming the stakeholders with

uncertainty (Jack et al., 2020). These scenario developments usually produce explorative or normative scenarios (Table A1) and need the (uncertain) climate change information as boundary conditions. To support local climate change adaptation, distributions of potential future climate change hazards from global-scale multi-model ensembles can be integrated with local data in Bayesian Networks that represent the causalities of , e.g., local water scarcity (Kneier et al., 2023). Then, adaptation strategies should be developed, which work under multiple plausible futures, i.e., which are more robust and adaptive (Maier et al., 2016), and incorporate the acceptance of the relevant actors to implement the adaptation measure(s). In the KlimaRhön project, this was done in the workshops that followed the first workshop, in which the climate change hazards, to which the stakeholders wanted to adapt, were identified.

## 5 Conclusions

With ongoing climate change, adaptation to climate change has to happen everywhere around the globe at local to regional scales. Adaptation measures should be identified in participatory processes involving local stakeholders and professionals with a scientific background, by embracing the multiple uncertainties that affect the future success of adaptation measures. In this paper, we present a readily applicable approach for quantifying and communicating climate change hazards and their uncertainties with multi-model ensembles. This approach is applicable in many climate change adaptation processes worldwide; it is not restricted to hydrological hazards but can also be used in climate change adaptation processes in the fields of agriculture, forestry, fisheries, and biodiversity.

The presented method for producing quantitative estimates of future climate change hazards, which benefits from the freely available output of global multi-model ensembles (provided by the ISIMIP initiative), can be replicated by anybody with basic knowledge in any programming language such as R, Python, or MatLab. Due to the high uncertainty of the translation of climatic changes into hydrological changes, utilization of the multi-model ensemble output is preferable even for local study areas unless multiple local hydrological models are available; with only one hydrological model, the uncertainty of future changes would be underestimated. We recommend quantifying hazards as relative changes as these can be estimated more robustly by multi-model ensembles than absolute values or changes of absolute values.

Based on our experience, we recommend using our approach to use different uncertainty visualization formats to communicate the range of potential future changes to either stakeholders in a climate change adaptation process or the general public. Stakeholders who need to identify adaptation measures based on uncertain futures hazards are best informed about the hazards by percentile boxes that show which relative change of a variable is exceeded according to which percent of all ensemble members. Distinguishing five percentiles in an easy-to-grasp visualization with an appropriate degree of complexity, percentile boxes enable the stakeholder to select to which future changes they plan to adapt depending on their risk aversion. For the presentation of climate change hazards to the general public, a simple table with the mean changes and an indication of the agreement of the models on the sign of change is preferable. Communicators should always reflect and decide what information should be the focus of a visualization.

When presenting climate change hazards, we propose to communicate what share of the multi-model ensemble simulates a change instead of stating this share of the multi-model ensemble as a probability. This communication approach avoids the uncertain relation of ensemble percentiles to probabilities and moves the multi-model ensemble from a shallow to a shallow medium uncertainty level. We suggest that an improved visualization and communication format for the important changes in interannual variability is investigated in the future.

However, as legitimacy, credibility and salience are perceived differently by individual stakeholders, no perfect, standard method to communicate information can be identified; "our worldviews, values and social norms dictate how we receive information and apply it" (Corner et al., 2018, p. 3). A potential remedy is to implement the Cultural Theory into the communication strategy, which categorizes people into four cultural world views when dealing with risks: hierarchists prefer expert opinions and regulations, egalitarians value societal contribution for risk reduction, individualists prefer market-based solutions and fatalists are apathetic viewing risks as unpredictable and random (Verweij et al., 2006; Czymai, 2023). These cultural worldviews could be integrated into the communication strategy to convince a heterogeneous audience to embrace uncertainty in their decision-making and the impact should be evaluated. To address hierarchists, it could be communicated that practitioners and scientists view uncertainty information as relevant (Höllermann and Evers, 2017). For egalitarians, it could be communicated that embracing uncertainty promotes fairness and prevents exposing only a few individuals to hazards due to collective inaction. Individualists could be approached by elucidating that embracing uncertainty could maintain their capacity to act and foster innovation.

Despite the coarse model resolution and wide uncertainty ranges, the multi-model ensemble results and their suitable communication helped the stakeholders in the participatory KlimaRhön process to understand uncertainty and to develop robust and flexible adaptation options. With our approach to quantifying and communicating multi-model ensemble results as a basis, flexible climate change risk management strategies can be developed jointly by stakeholders and scientists in a participatory and transdisciplinary manner.

**Appendix**

**A1 Definition of central terms**

**Table A1. Definition of central ambiguous terms that can be used in the communication of climate change risks.**

| Term | Description |
| --- | --- |
| Risk | "Disaster risk signifies the possibility of adverse effects in the future. It derives from the interaction of social and environmental processes, from the combination of physical hazards and the vulnerabilities of exposed elements […]." (Cardona et al., 2012, p. 69). |
| Hazard | "[…], hazard refers to the possible, future occurrence of natural or human-induced physical events that may have adverse effects on vulnerable and exposed elements […]. Although, at times, hazard has been ascribed |

| | |
|---|---|
| | the same meaning as risk, currently it is widely accepted that it is a component of risk and not risk itself." (Cardona et al., 2012, p. 69). |
| Uncertainty | Uncertainty means that we have limited knowledge about something (Marchau et al., 2019). Mahmoud et al. (2009, p. 806) state that "[u]ncertainty is the inability to determine the true magnitude or form of variables or characteristics of a system […]". According to Döll and Romero-Lankao (2017), uncertainty in participatory climate change risk management has three dimensions, which are position, nature and level of uncertainty: <br><br> • The position of uncertainty indicates, in which part of the participatory process uncertainty appears. <br><br> • The nature of uncertainty can be epistemic (either substantive or in the participatory process), ontological, ambiguous, or linguistic. <br><br> • Uncertainty can be classified in levels or degrees from shallow, medium over deep uncertainty to recognized ignorance. <br><br> For more details, see Döll and Romero-Lankao (2017, Figure 1 and pp. 22-24). <br><br> The uncertainty considered in this study, which is embraced by multi-model ensembles, can be categorized as shallow to medium. |
| Scenario | Scenarios describe alternative, plausible future developments (Maier et al., 2016). They are developed to address deep uncertainties (Börjeson et al., 2006; Döll and Romero-Lankao, 2017; Voinov et al., 2018), i.e., when the uncertainties cannot be described as probability distributions (Maier et al., 2016). |

**A2 Impact models**

Quantification of potential future changes of relevant characteristics of the physical system such as groundwater recharge or agricultural yield is achieved by analyzing either the output of climate models or the output of impact models that are driven by the output of climate models. Impact models are domain-specific models such as hydrological models or crop models that focus on the simulation of certain subsystems of the Earth system. Analysis of impact model output is preferred if climate models do not compute the variable of interest or if, e.g., due to the spatial scale or the bias of climate model output, impact

models can be assumed to provide a better representation of reality. Due to the low accuracy of climate models when simulating current climate conditions, climate model output is first bias-adjusted using historic climate data before it is applied to drive impact models (Frieler et al., 2017).

**A3 Multi-model ensemble**

The multi-model ensemble of the study consists of 32 model combinations of four global climate models and eight global

hydrological models, which took part in the ISIMIP2b project. The four global climate models are IPSL-CM5A-LR, GFDL-ESM2m, MIROC5 and HadGEM2-ES. The eight global hydrological models that were used are CLM4.5, CWatM, H08, JULES-W1, LPJmL, MATSIRO, PCR-GLOBWB and WaterGAP2. In the ISIMIP2b project, ISIMIP prescribed in its protocol

(ISIMIP, 2019) that the models should be run with different climate and $CO_2$ concentration scenarios and socio-economic scenarios. For the future periods, we used the model output of the $CO_2$ concentration scenarios "rcp26" and "rcp85" and each with the socio-economic scenario "2005soc". For the reference period, we used the socio-economic scenario "histsoc" with the $CO_2$ concentration scenario "historical".

However, some models cannot execute some of these runs. The global hydrological model PCR-GLOBWB was not run for the climate and $CO_2$ concentration scenario "rcp85" which is why the multi-model ensemble only consists of 28 model combinations for the scenario "rcp85". Moreover, the hydrological model CLM4.5 was not run with the socio-economic scenario "histsoc" but with the socio-economic scenario "2005soc" in the reference period, and the global hydrological model JULES-W1 is not run with any of the previously mentioned socio-economic scenarios but with the scenario "nosoc".

**A4 ISIMIP**

ISIMIP developed a protocol with which international global hydrological modelling groups consistently executed the simulations (Frieler et al., 2017), resulting in simulation outputs available for download as NetCDF files (see https://data.isimip.org/). According to the protocol, each impact model was driven by the same bias-corrected output of four global climate models (GCM), where each GCM was driven by four emissions scenarios or rather representative concentration pathways (RCP) (Frieler et al., 2017). Each impact model that takes part in ISIMIP follows the same simulation protocol (Frieler et al., 2017), which ensures that the modelling results are comparable and can therefore be included in a multi-model ensemble. With the different emissions scenarios, the deep uncertainty of future anthropogenic emissions is considered. Experts can use the freely available ISIMIP model output to perform multi-model ensemble analyses for different sectors (Warszawski et al., 2014) and thus characterize the uncertainties of future changes in impacted variables. Looking across each ISIMP multi-model ensemble, the projected change of each ensemble member should be assumed to be equally likely. Please note that the ISIMIP multi-model ensembles do not cover the whole range of uncertainty, one reason being the low number of utilized global climate models.

**A5 Future period**

The "far future around 2084" had to be shifted by one year compared to Hübener et al. (2017) due to the ISIMIP 2b data being only available until 2099.

**A6 Greenhouse gas emissions scenarios**

Scenario RCP2.6 describes a pathway with strong mitigation measures resulting in global warming that likely will not exceed 2 °C until the end of the 21st century (compared to the "pre-industrial" period 1850-1900) and therefore meets the goal of the Paris Agreement (Collins et al., 2013). Under RCP8.5, high emissions of greenhouse gases are assumed that lead to global warming of approximately 4 °C until the end of the 21st century (Collins et al., 2013). The intermediate RCPs RCP4.5 and

RCP6.0, which are available in ISIMIP 2b, were not used in this study to not overwhelm the stakeholders; the chosen RCPs 2.6 and 8.5 are to show the range of outcomes of the best-case and the worst-case climate futures.

## A7 Area-weighted averages

Four of the 0.5° grid cells of the global hydrological models overlie the whole study area of about 2,433 km², covering 506 km², 532 km², 829 km² and 566 km², respectively (Figure 2). We calculated area-weighted averages for the BRR area from the four grid cell values and only used this average in our hazard quantification as global hydrological models cannot reliably quantify differences between the four grid cells and thus sub-regions of the BRR.

## A8 Interannual variability

The exceedance probabilities were calculated with the annual averages as well as the seasonal averages of the summer and winter months. We did not analyze floods as GHMs are not suitable for simulating local floods. We assessed droughts only by considering the change in statistical low-flow values of total runoff (particularly relevant for stream ecosystems) and groundwater recharge both for annual and summer values (see above).

## A9 Relative changes of long-term mean changes

We calculated the potential relative changes of both groundwater recharge and total runoff between the reference period and the two future 30-year periods – the "near future around 2035" and the "far future around 2084" (Section 2.1.2). As indicators of the hydrological hazard of climate change, the changes between the 30-year periods were computed as relative changes for each GCM-GHM combination individually. For the interannual variability, we calculated the relative changes between the values of the same rank in the future and the reference period resulting in changes in exceedance probabilities.

## A10 Interpretation of 30-year mean change

For the annual values (grey percentile boxes), around 50% of the multi-model ensemble simulate a decrease and around 50% an increase of total runoff and groundwater recharge in both future periods. For the seasonal values of total runoff, also around half of the multi-model ensemble simulates a decrease and around half an increase for the near future around 2035, while a slight majority of the models predict an increase in the summer and winter months for the far future around 2084 (suggesting decreases in spring and fall).

Regarding seasonal changes of groundwater recharge, at least 70% of the multi-model ensemble simulate an increase of groundwater recharge during the winter months in both future periods and under both emissions scenarios. In contrast, at least 70% of the multi-model ensemble simulate a decrease of groundwater recharge during the summer months in the near future around 2035, and in the far future around 2084 under the emissions scenario RCP 8.5. In the near future around 2035, 10% of the model ensemble simulate a decrease in groundwater recharge in summer months of more than 50%, while 10% of the model ensemble simulate an increase in groundwater recharge in winter months of more than 65%. In the far future around

2084 under the emissions scenario RCP 8.5, 10% of the model ensemble simulates a decrease of more than 60% in the summer months and an increase of more than 100% in the winter months. For summer months in the far future around 2084 under the emissions scenario RCP 2.6, the median change is close to zero but 10% of the model ensemble projects a decrease of more than 50%.

## A11 Interpretation of interannual change

We assessed the changes of total runoff and groundwater recharge for all months as well as only the winter and only the summer months of each year. In the case of RCP 2.6, around 50% of the multi-model ensemble simulate a decrease and around 50% an increase of groundwater recharge in years with relatively high summer groundwater recharge (left side of the right diagram in Figure 6). The ensemble median (green solid line) drops to the right, and potential changes of groundwater recharge in those years with a relatively low summer recharge (from the exceedance probability of 70%), P70 (upper, green dashed line) drops below 0%. This means that 70% of the multi-model ensemble simulate a decrease of groundwater recharge in case of relatively dry summers. In the case of RCP 8.5, at least 70% of the multi-model ensemble simulate a decrease of groundwater recharge in the far future around 2084 under the emissions scenario RCP 8.5 in most wet and dry years. The ensemble median decrease of groundwater recharge becomes larger from wet to dry years, and 10% of the ensemble members project that the summer recharge in wet years (10% exceedance probability) will decrease by at least 45% but summer recharge in dry years (90% exceedance probability) will decrease by at least 90%. However, this behavior needs to be interpreted carefully because relative changes are higher, for the same absolute change (in mm), when absolute values are small (Betts et al., 2018).

## Data Availability Statement

The map in Figure 2 was created with ArcMap 10.8.1 by Esri, available with a license at https://www.esri.com/en-us/home. The data used for Figures 5, 6 and 10 are from the ISIMIP project, following the ISIMIP2b simulation protocol (ISIMIP, 2019), and can be accessed at https://www.isimip.org (last access: 27[th] of October 2022) (Frieler et al., 2017). The data was processed and analyzed with Python (https://www.python.org/) in the Python integrated development environment PyCharm (available with a license at https://www.jetbrains.com/pycharm/). The script and an example data set to generate a comparative figure of uncertainty visualization formats for the same data (Figure 10) was made freely available (Müller, 2023). The evaluations of workshop 1 was realized with SoSci Survey (Leiner, 2019) and was shared with the stakeholders via www.soscisurvey.de. The evaluation results of the first workshop in the climate change adaptation process, which are shown in Figure 8, are freely available in Zenodo (see Müller and Czymai, 2022; Creative Commons Attribution 4.0 International). The evaluation of the in-person presentation was realized with printouts and that of the online presentation was realized in the webinar software edudip (https://www.edudip.com/) and downloaded as a PDF file. The evaluation results were digitalized and analysed with Microsoft Excel (Figures 8 and 9) and Figures 1 and 4 as well as the slides shown in Figures 3 and 7 were created in Microsoft PowerPoint both available with a license at https://www.microsoft.com/de-de/microsoft-365.

**Authors' Contributions**

Laura Müller conducted the conceptualization, analysis, methodology, visualization, and writing of the original manuscript. Petra Döll had the original idea for the manuscript, supported and supervised all steps and reviewed the manuscript.

**Competing interests**

The authors declare that they have no conflict of interest.

**Ethical statement**

The evaluations were carried out based on voluntary participation and informed consent to publish the outcomes. The evaluation data was gathered and saved in an anonymous manner. For this, the stakeholders were asked to evaluate (among others) the communication format in an online evaluation at the end of the digital workshop 1 and they were informed before that the evaluation results may be published anonymously. In the in-person presentation outside of the participatory process, consent documents were collected from each participant regarding their voluntary participation and their agreement to publish the evaluation results anonymously. In the online presentation outside of the participatory process, the audience was asked to evaluate the communication formats in an online evaluation and they were informed that the evaluation results may be published anonymously. The research conducted in this study is novel, representing the authors' perspectives, and does not involve any intent to cause harm to others.

**Acknowledgments**

We thank Birgit Blättel-Mink and Max Czymai with whom we organized and conducted the workshops in the KlimaRhön project. We are grateful for the support of the three administrations of the UNESCO biosphere reserve Rhön, who were the key stakeholders of the KlimaRhön project, and thank all the stakeholders who participated in the workshops for their engagement in the discussions. Moreover, we would like to thank the ISIMIP project (www.isimip.org) and Hannes Müller Schmied for providing the model data. We acknowledge the Hessian Agency for Nature Conservation, Environment and Geology (HLNUG) and its Centre on Climate Change and Adaptation (FZK) (https://www.hlnug.de/themen/klimawandel-und-anpassung) for funding the KlimaRhön project. We are very thankful for the thorough comments of the reviewer Usha Harris and an anomymous reviewer that helped to improve the manuscript. We acknowledge the use of ChatGPT for generating sentence examples and translations of specific words, contributing to the linguistic comprehensibility of our work.

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
