# Peer review of "Quantifying and Communicating Uncertain Climate Change Hazards in Participatory Climate Change Adaptation Processes"

_EGUsphere, 2023_

## Author Comment (AC2)

Preprint egusphere-2023-1958 (https://doi.org/10.5194/egusphere-2023-1958)

Response to Anonymous Referee #2

Dear reviewer,

Thank you very much, we are grateful for your constructive, detailed, and in-depth suggestions and helpful comments for improving our manuscript. Below, each reviewer's comment (indicated by "RC") is followed by our answer (indicated by "AC"). The proposed new text in the revised manuscript is written in bold.

**RC:** The paper aims to present an approach for quantifying and communicating climate change-related uncertainties and outputs of models to stakeholders as part of a participatory climate change adaptation process. The paper provides information about the context and the participatory process used in this study, the variables used, and how results are communicated and perceived by the audience, and it discusses issues around communicating uncertainty.

The topic of the manuscript is helpful for scientists who want to identify ways to better communicate uncertainty to stakeholders, such as the uncertainty relating to climate change. The adopted approach and the study's outcomes provide insights into how communication can shape stakeholders' understanding and subsequently influence decision-making on adaptation strategies. In addition, the study provides valuable information on aspects that should be considered when designing content to visualise uncertainty.

**AC:** Thank you for the positive feedback.

**RC:** However, the manuscript should be improved to bring out the valuable points it discusses in a more structured manner, increasing the impact of the study and broadening the readership.

First, as a general comment, the paper would benefit from some serious editing; the text is too long, and the story gets easily lost. Providing the information more concisely will make it a more valuable contribution. Specific suggestions on how the manuscript can be improved are included below.

**AC:** Later, we will show concrete ideas where and how we will make the text more concise.

**RC:** Specific comments

The paper's objective relates to two distinct tasks: (a) how to assess climate change hazards with their uncertainties from multi-model outputs and (b) how to communicate this information in a climate adaptation-focused participatory process. However, considering how the information is presented in the paper sections, it is unclear how these tasks are addressed and what the key messages are. It might help readers to follow the study more effectively if the structure of the paper was slightly re-organised. As a suggestion, the authors could distinguish the two tasks and provide details on the approach and results for each. For example, in chapter 3, where the adopted approach is presented, there could be two different clearly labelled sections, one for each objective, providing details on the methodology used to address these. Similarly, in chapter 5, the discussion could address the two distinct tasks more explicitly.

**AC:** As proposed, we will restructure our paper with the two foci (a) and (b) to highlight the main objectives of the paper. This is how we will structure the paper in the revised version:

1 Introduction
2 Quantifying and communicating the uncertainty of the climate change-induced hydrological hazards
2.1 Quantification of hydrological hazard indicators

2.1.1 Study area
2.1.2 Processing of multi-model ensemble output
2.2 Communication of climate change hazards
2.2.1 Participatory process
2.2.2 Communication of the quantification
The heading of former 3.3.1 will be dropped; the text will be directly under the body text
2.2.2.1 Communication of potential changes of 30-year mean values by percentile boxes
2.2.2.2 Communication of potential changes in interannual variability
2.2.2.3 Summarizing hazards for the stakeholder discussion
2.2.2.4 Alternative communication of potential changes of 30-year mean values by tables
3 Results
3.1 Interpretation of communicated hazard indicators by the stakeholders (last paragraph of formerly 3.3.4)
3.2 Evaluation of communication format by the stakeholders of the KlimaRhön participatory process
3.3 Comparison of our communication format with a more common communication format by the audiences of two presentations of the project results
4 Discussion
4.1 Why and how should the uncertainty of hydrological changes due to climate change be quantified with a multi-model ensemble of global models?
4.2 How can the uncertainty of hydrological changes due to climate change be best communicated?
4.3 Evaluations in participatory processes
4.4 Using the uncertain information about future climate change hazards for the development of adaptation measures
5 Conclusions

**RC:** The paper includes an abstract and a plain-language summary; however, there is no significant difference between these two sections. From a typical reader's perspective and if the authors consider the need for a plain language summary, I suggest focusing on the problem this study tries to solve and why this study is important and reference the adopted approach and results using simple language to allow an average reader to understand. In addition, making reference to the usability of such an approach in a different context would also add value. In the case of the abstract, and aiming to help the readers identify the essence of the study and remember the key points, the authors could consider adding details to clarify the study's objectives and overall contribution.

**AC:** In the revised version, we will highlight that our approach can be applied anywhere around the world due to the global coverage of the freely available data of potential future hydrological changes and also in other contexts. Moreover, we will structure it more clearly after the two main objectives (a) and (b). In the Plain Language Summary, we aimed to have the (almost) same content as in the Abstract but in easier words, i.e. outlining what the paper is about. We would not write about another focus, because this might set other expectations of the reader.

"Abstract. Participatory processes for identifying local climate change adaptation measures have to be performed all around the globe. **As these processes require information about context-specific climate change hazards, knowledge about how to quantify climate change hazards and how to best communicate the potential hazards with their uncertainties is essential.** In a participatory process on water-related adaptation in a biosphere reserve in Germany, we used the freely available output of a multi-model ensemble **provided by the ISIMIP initiative, which provides global coverage,** to quantify the wide range of potential future changes in (ground)water resources. **Our approach for quantifying the range of potential climate change hazards can be applied worldwide even for local study areas, and also for adaptation in agriculture, forestry, fisheries, and biodiversity**. To support participatory climate change adaptation processes, we propose to **communicate** uncertain local climate change hazards with percentile boxes rather than with boxplots or simple average **with the** model agreement on the sign of change. This **supports the** stakeholders in identifying the future changes they wish to adapt to depending on the problem (e.g., resource scarcity vs. resource excess) and their risk aversion. Using or adapting our quantification and communication

approach, flexible climate change risk management strategies can and should be developed worldwide in a participatory and transdisciplinary manner involving stakeholders and scientists."

**RC:** The introduction section seems too long and makes it difficult for the average reader to understand the background, context, and problem this study aims to solve. Also, how it is written makes it difficult for the reader to identify the gap in the literature this paper addresses. The suggestions below can improve the contextualisation of the study and will help the reader follow the next sections better:

- **RC:** Provide information on existing approaches; this could be a table presenting advantages and disadvantages and highlighting challenges when communicating uncertainties. The proposed table will allow the reader to understand the existing approaches, what is missing, and what this paper seeks to address.
  **AC:** In the discussion (new section 4.2), we will introduce Table X explaining the advantages and disadvantages of uncertainty visualization to communicate uncertainty to stakeholders. Moreover, we will explain why we selected our visualization method, which is a modified visualization of the GERICS bar chart.

Table X: Advantages and disadvantages of some uncertainty visualization to communicate uncertainty to stakeholders.

| Uncertainty visualization | Advantages | Disadvantages |
|---|---|---|
| Boxplot | Shows three quartiles showing the change values that are not exceeded by a certain percentage of the ensemble members; defines outliers; common visualization | Potentially reduced readability as it shows minimum and maximum; non-unique definition of and difficult to interpret "whiskers" and, thus, outliers; does not show the distribution of values |
| GERICS bar chart | Shows five percentiles P0, P20, P50, P80, and P100 showing the change values that are not exceeded by a certain percentage of the ensemble members | Potentially reduced readability as it shows minimum and maximum; does not show the distribution of values |
| Violin plot | Very precisely visualizes uncertainty by displaying the distribution of values within the whole range by the width of the box | Potentially reduced readability as it shows minimum and maximum; only shows the percentiles minimum, maximum, and median; smoothing of the shape leads to only an approximate representation of the distribution; the high information content due to the high uncertainty precision might be overwhelming for the end user |
| Letter Value plot | Shows several percentiles showing the change values that are not exceeded by a certain percentage of the ensemble members | Potentially reduced readability as it shows minimum and maximum; area of the Boxes does not correspond to the number of values contained in the boxes; pseudo- (i.e. misleading) visualization of the distribution of values; depending on the visualization, it has too many percentiles |
| Percentile box | Shows five percentiles showing the change | Does not show the distribution of values |

| | values that are not exceeded by a certain percentage of the ensemble members; percentiles can be selected depending on the problem and the risk aversion; minimum and maximum can be avoided preventing the misinterpretation that these are the most extreme values possible | |
|---|---|---|

- **RC:** There is information in other sections of the manuscript that can be used in the Introduction section to improve the contextualisation of the study. For example, in section 5 (§5.1.1 and 5.1.2), information around uncertainty and available models and visualisation formats could be used to set the background, preparing the reader about what is coming up and what this study is trying to achieve. Similarly, information included in section 6 around the theoretical background (1st paragraph on page 29 - lines 741-752) could be used to shape the contextualisation of the study.
  **AC:** We will structure the introduction in a way, which highlights the main objectives (a) and (b) more. For this, we will introduce one sentence, which outlines the structure of the introduction in line 51 after the second paragraph "**This is why (a) future changes should be assessed with their uncertainty and (b) a suitable visualization should be found with which the future changes with their uncertainty can be communicated.**"
  In line 102, we will add "**To communicate the processed potential changes with their uncertainties visually, a suitable visualization format is needed** and should not be a translation into median or mean changes only, […].".Then, in the following paragraph, we will include information from the former 5.1.1 and 5.1.2. For this, we will move lines 553-558 in the Introduction. Moreover, as in the answer to the previous comment, we will move the paragraph in lines 579-587 to the methods section.
  In line 118, after the sentence, we will introduce:
  "**Similarly, the letter value plot shows several percentiles with bars, but with reduced bar width the more distant it is to the median (Figure 10). To show the distribution of values, violin plots can be used, which also show the minimum, median, and maximum values (Figure 10).**"
  As we have not embraced the Cultural Theory in the communication of the uncertainties in the participatory process, we will not move the 1st paragraph on page 29 (lines 741-752) from the conclusion to the introduction but leave it in the conclusion as a recommendation.
- **RC:** To reduce the length of this section and improve its readability, I suggest making the presented information more concise and reducing repetition. Removing the 2nd paragraph on page 3 (lines 76-89) and adding the content as supplementary material would help reduce the length and allow the typical reader to focus on the context of the study.
  **AC:** We find the paragraph on ISIMIP very valuable for persons who seek to analyze data as we did for another sector to motivate that our approach is also possible with other than hydrological data. But, in the revised version, we will shorten the 2nd paragraph on page 3 as shown:

  "**The Inter-Sectoral Impact Model Intercomparison Project (ISIMIP, www.isimip.org) provides freely available multi-model ensembles of many impact variables in several impact sectors (water, lakes, biomes, regional forests, permafrost, agriculture (crop modelling), energy, health, coastal systems, fisheries and marine ecosystems, and**

**terrestrial biodiversity; ISIMIP, 2019). For each impact variable, ISIMIP2b provides a time series for historic and future periods, which were computed by multiple global impact models (Frieler et al., 2017), with which the uncertainties of future changes in impacted variables can be characterized.**"

and integrate it more with the explanation of multi-model ensembles. The rest of the paragraph will be moved to the Appendix in the revised version.

We will move the explanation of models in lines 52-59 to the Appendix and integrate the last sentence of the paragraph somewhere else in the revised version.

**RC:** Section 2 provides information about the case study and the participatory process. The authors could consider if this section could be incorporated into the methodology and results chapters. For example, the case study part §2.1 presents the study area and, therefore, could be part of section 4, which presents the study results. Also, the participatory process part §2.2 discusses the approach adopted in this study and links to the methodological aspects. As a suggestion, this part can be included in Section 3, which discusses the overall methodology. In this way, the information will be presented in a more structured way, allowing an average reader to follow the paper better.

**AC:** In the revised version, we will move both Sections in the former Chapter 3 as outlined above.

**RC:** Section 3 presents the approach adopted in the study. This section seems too long and makes it difficult for an average reader to understand the approach and reasoning behind it. I suggest the following changes that will improve the way the information is captured and will enable the reader to understand easily the adopted process, broadening the readership of the manuscript:

- **RC:** The authors could explain the methodology used per objectives (a and b) as mentioned previously; there could be two different clearly labelled sections, one for each objective, providing details on the methodology used to address these. Also, to allow readers to follow the logic of the methodology more closely, the authors could consider including a graphical abstract of the main steps of their approach per objective.
  **AC:** As shown previously, we will change the structure of this Section in the revised version. A graphical abstract (Figure X) will be included as Figure 2 at the beginning of the former Section 3 "Quantifying and communicating the uncertainty of the climate change-induced hydrological hazards" (Section 2 in the new structure).

[Figure]

Figure X: Schematic of the presented approach of quantifying and communicating uncertain climate change hazards in participatory climate change adaptation processes. ISIMIP: The Inter-Sectoral Impact Model Intercomparison Project (www.isimip.org/).

- **RC:** Sections 3.1 and 3.2 provide information on the hydrological hazard indicators; although the information is useful, it can distract the reader from understanding the adopted approach. I suggest the authors provide a brief summary in this section and include the information as supplementary material to improve the readability.
  **AC:** In the revised version, the former sections 3.1 and 3.2 will be merged (see above). The text will not be structured by subsubsections anymore and will be made more concise (very technical details important to applying the approach will be moved to the Appendix). Thus, the first two sentences of the paragraph "Greenhouse gas emissions scenarios" (the former 3.1.2) will stay in the text, the rest will be moved to the Appendix.
- **RC:** Sections 3.1 and 3.2 provide information on the analysis conducted, addressing the objective (a) of the study; what is missing to allow readers to follow the approach more closely is the output of this task, what this analysis provides and how it feeds into the next step (objective b). Currently, this information is captured in section 3.3; for example, the analysis output was the design of graphics used in the participatory process (as mentioned in §3.3.2, line 325).
  **AC:** To make the connection between the objectives (a) and (b), we will rewrite the first paragraph of the former Chapter 3:
  "**Future changes should be assessed with their uncertainty and then a suitable visualization should be found with which the future changes with their uncertainty can be communicated in participatory processes. For this,** scientists or experts have to decide on what and how to produce climate change risk information before they communicate it to local stakeholders.  So, during the first step, the data processing **and the analysis,** they need to decide what indicators of climate change hazard should be quantified, given the problem, the interest of the stakeholders, data availability and quality as well as technical and time constraints. **In the second step,** the scientists, experts or communicators have to decide on what, **with which visualization format** and how to communicate given their audience, the aim and the generally severe time constraints in the participatory process."
- **RC:** Section 3.3 provides information on communicating the hazard indicators, so it mainly addresses the objective (b) of the study. Although the section provides detailed information on the process followed, how the information is presented can be confusing for an average reader. The authors could consider (as mentioned previously) including the participatory process part §2.2 in this section, as it addresses methodological aspects adopted in this study. The overall process and information could be presented in a graph or a table, showing details on the workshops, timeline, participants, objectives, and what has been achieved. Also, the authors could consider adding information to reflect why they chose the specific way of communication to allow the reader to understand the driver behind the specific approach. Furthermore, information relevant to the options provided to stakeholders (mentioned in section 4, page 21) could also be included here. This will improve the paper's readability, allowing the reader to follow the logic of the participatory process more closely.
  **AC:** As mentioned above and proposed by the referee, we will move the Section about the participatory process to the communication section. Moreover, we will move lines 505-511 and Figure 8 to the former Chapter 3 (see above). Together with the table of the advantages and disadvantages of uncertainty visualizations, we will include two sentences reflecting why we have not chosen existing uncertainty visualizations in the Discussion (in section "4.2 How can the uncertainty of hydrological changes due to climate change be best communicated?"). However, we think that more information on the participatory process will not enhance understanding as the paper only covers the very first workshop in the participatory process.
- **RC:** Section 3.3 is too long and includes details that can confuse the average reader. To improve the readability, I suggest the authors consider including information that is relevant to the approach only and not the results and presenting it more concisely. For example, the last paragraph of the section on page 18 (lines 442-454) presents some of the results that can be included in section 4 (results).
  **AC:** In the revised version 2.2.2.3 "Summarizing hazards for the stakeholder discussion" (former 3.3.4) will be reduced by the second paragraph, which will be moved to 3.1 "Interpretation of communicated hazard indicators by the stakeholders".

- **RC:** The authors could consider adding a more detailed description in Figure 2 (page 13) to allow the reader to understand why the selected multi-model ensemble represents the currently best estimate of future hydrological hazards.
  **AC:** Figure 2 shows exactly the slide that we showed and explained to the stakeholders. We wanted to show the reader exactly what we showed the stakeholders, and therefore do not want to add anything to Figure. 2. Later, in (former) Section 5, we discuss the advantages and disadvantages of the chosen multi-model ensemble.
- **RC:** Average readers might be able to follow the graphs presented in Figure 4 (page 15) more clearly if there was a more thorough explanation of the potential change of groundwater recharge in the caption, as mentioned on page 14 (lines 345-352). A similar approach could be adopted for Figure 5).
  **AC:** It is not the main intention of the article to present our interpretation of the processed potential changes but to show how we communicated the potential changes and how we communicated our interpretation of the potential changes. Therefore, we find it too exhaustive to add the interpretation in the figure caption. But we will refer to lines 345-352 in the caption of Figure 4 and lines 398-404 in the caption of Figure 5 in the revised version.
- **RC:** In section 3.3, pages 14-15, lines 359-368, information more relevant to the analysis rather than the communication aspects is included. The authors could consider whether this information can be moved to section 3.1 (as per the previous comment) to allow readers to follow the logic of the methodology more closely.
  **AC:** In the mentioned paragraph, the processed data is interpreted (also in the paragraph in lines 405-416), which is neither part of our communication approach nor part of our quantification approach. We will move these two parts in the Appendix and will refer to them in the captions of Figures 4 and 5 respectively, which is not the perfect solution because the interpretation should be close to the visualization of the results for better comprehension.
- **RC:** To reduce the length of this section, terms and concepts (including Table 1) could be included in the introduction or supplementary material.
  **AC:** In the revised version, we will move Table 1 into the Appendix.

**RC:** Section 4 presents the results of the study. As mentioned in a previous comment, this section could include information about the case study (currently §2.1) at the beginning and then present the results.

**AC:** In the revised version, we will move the former §2.1 before §2.1.2 Processing of MME output in the revised structure.

**RC:** Figures 6 and 7 could be presented on the same page, allowing the average reader to compare the results easily.

**AC:** To make sure that Figures 6 and 7 will be presented on the same page, we will rearrange them and display them in one Figure (see below).

[Figure]

**RC:** The authors could consider adding information on the importance of the results and how these are dependent on the number of participants in the workshops. For example, based on the online participation, the results seem more significant than those from the in-person presentation. There is no information to determine whether the type of participants in the in-person workshop adds value to the results. Also, no information is included regarding the conducted statistical analysis of the results.

**AC:** The results of the evaluation by the stakeholders in the participatory processes is more important than those of the in-person presentations as we wanted to study methods for participatory processes on climate change adaptation. This has already been expressed in lines 650-652, 687-689 and in section 5.2 (of the preprint) in general. As we wanted to address all types of persons with our uncertainty communication, we did not make differences between the types of participants. We only discuss later in Section 5 that the information needs and the intention of listening to the uncertainty communication (interest in the presentation of project results vs. participation in a workshop to develop adaptation measures) is important for the suitability of uncertainty communication. Due to the low number of respondents, no further statistical analysis is deemed to be necessary.

**RC:** Finally, the authors could consider a different type of format for Figure 9 (perhaps combining the two visualisations in one) to avoid confusion with the graphs used in the participatory process and help the reader interpret the results. In addition, and as per the GC guidelines, the authors should use in the figures a colour combination that would allow readers with colour vision deficiencies to interpret the findings correctly.

**AC:** To avoid confusion, we will extend the figure captions of Figs. 7 and 9 to clarify that these evaluations refer to two presentations of the project results to persons that have not been involved in the participatory process. We will also add as titles in Figure 9 and the combined Figure of Figures 6 and 7: "Evaluation by audience of presentation of project results" and to the combined Figure also: "Evaluation by participants of the climate change adaptation process" (see Figures below and above).

In the revised version, we will also change the color combination for better visual readability (see Figure below). The color combination is colorblind safe and print friendly concerning https://colorbrewer2.org/.

[Figure]

**RC:** The discussion chapter (Section 5) could be further improved by adding more detail on interpreting the results in the context of the objective of this study rather than referencing the results, which is already done in Section 4. For example, considering the outcome of the workshops, did the authors further explore the stakeholders' views on the communication format that would make more sense to them?

**AC:** Within the scope of this paper, we did not explore stakeholders' views in addition to the evaluations that we presented in this paper. The aim of the whole participatory process (in which this

presented study just represented an aspect in the first of five workshops) was to identify adaptation measures in water management for the study area. We aimed to ensure that stakeholders did not perceive themselves as experimental subjects and recognized that their identification of measures was taken seriously, not merely as an observation of communication methods.

**RC:** How did the authors interpret the results considering the overall context of the study?

**AC:** In the revised version, we will include in 5.3 that the specific results of the multi-model ensemble were not used quantitatively (only qualitatively) in the discussion of adaptation measures because no technical measures were discussed or monetary cost-benefit analyses were performed.

**RC:** What are the key messages?

**AC:** We will include the following information in the conclusion of the revised version (see below):

Concerning objective (a):
To assess uncertainties with multi-model ensembles it is more robust using changes (not absolute values). Moreover, our approach is preferable to use even for local study areas except when another (local) multi-model ensemble is available, which is very rarely the case. The important changes of interannual variability were hard to grasp for non-scientists, thus, for their communication, another communication format is needed.

Concerning objective (b):
Based on our experience and the results, we believe that for stakeholders who need to identify climate change adaptation measures hazard communication by percentile boxes is preferable to communication by simple tables. For the presentation of climate change hazards to the general public, a simple table with the mean changes and an indication of the agreement of the models on the sign of change is preferable.

**RC:** What is the novelty of the approach, what are the advantages and disadvantages, and why was this specific approach chosen? What are the limitations?

**AC:** The novelty of our approach is that we highlighted that multi-model ensembles exist and that their results are globally available. We show how to make use of a global multi-model ensemble analysis for a small study area in very detail (processing can be done with basic knowledge in any programming language such as R, Python, or MatLab) so that it can also be done worldwide. This is especially useful for locations where no local multi-model ensemble is available. Moreover, in the paper, we show in very detail how uncertainty can be communicated with various uncertainty visualization formats in a real participatory process. We will combine these novelty aspects with the key messages and rewrite the Conclusions with the information (see below).

In the revised version of the Discussion, we will include in lines 595-596:

"The percentile box shows five (P10, P30, P50, P70, and P90) percentiles**, thus transparently visualizing uncertainty to stakeholders, enabling communication of more thresholds than other visualization formats and an advantage is that the percentiles can be chosen individually depending on the risk aversion and the problem. However, the boxplot only shows** three percentiles (P25, P50, and P75) **and includes** the difficult-to-interpret boxplot "whiskers" (Fig. 10)."

To stress the disadvantages, we will include in line 619:

"**However, the communication, i.e. the explanation of the approach and the interpretation, takes a lot of time in a workshop and asks the stakeholders for a long concentration span.**"

Moreover, in lines 579-624, we showed the advantages and disadvantages of our approach (the percentile box and the continuous percentile box) compared to other uncertainty visualization formats.

At the end of the first paragraph (line 552), we will include a technical limitation:

"**However, it needs an expert with basic technical knowledge in any programming language to assess the potential changes.**"

Before the second paragraph of the former §5.1.2, we will explain why we chose our approach:

"**We came up with our approach because we wanted to communicate more percentiles and did not want to communicate minimum and maximum values. We left out the minimum and maximum 1) to not give too much room to possible outliers and 2) to not give the impression that minimum and maximum values could not be exceeded in reality. We wanted to have several percentiles to have the possibility to say how many percent of models simulated a stronger or weaker change than a certain change value, which supports communication that does not include a specific prediction.**"

**RC:** Also, a typical reader might not understand how the specific approach adopted in this study tackled some of the issues mentioned in the chapter. For example, how did it address the transparency?

**AC:** In the last paragraph of (the former) 5.1 (lines 658-666), we will include that we showed the stakeholders the uncertainty of the models through multi-model ensemble and showed them the approach of our quantification to address the uncertainty routine "transparency".

"**By showing the stakeholders the uncertainty of the models through multi-model ensembles and showing them the approach of our quantification,** we address one of the uncertainty routines of stakeholders called "transparency", in which the stakeholder **considers** the limits of knowledge (Höllermann and Evers, 2019)."

**RC:** How did it help the stakeholders in decision-making? Explicitly exploring and answering these questions would lead to a much stronger paper with a more broadly applicable impact and will allow for broadening the readership of the manuscript beyond subject area experts.

**AC:** We cannot make reliable statements about how our approach helped the stakeholders in decision-making because we guess (and hope) that the whole participatory process influenced their decision-making and because we did not evaluate how their decision-making changed. We will integrate in the Discussion that we hope to have highlighted uncertainty enough so that they will more carefully look at uncertainties in their decision-making in the future. In line 703, we will include:

"**Due to highlighting the uncertainty of future changes, we hope that the stakeholders will more carefully embrace uncertainty in their decision-making in the future.** Next to the uncertainty of…"

**RC:** The conclusions section (Section 6) would benefit from a synthesis of the main points of the study, highlighting the advantages of the adopted approach and the importance of the paper more briefly. This would help the reader understand why the study should matter to them after having finished reading the paper.

**AC:** We will rewrite the Conclusions in the revised version integrating the key messages, novelty aspects, and advantages formulated above and structuring after the objectives (a) and (b).

Here is how we want to revise the Conclusions:

[revised manuscript text omitted]

RC: Finally, as a suggestion and according to the GC guidance, the ethical statement should be more comprehensive, and a description of the process should be included in the methodology section of the paper.

AC: We will add the following in line 791 to make the ethical statement more comprehensive:

"**For this, the stakeholders were asked to evaluate (among others) the communication format in an online evaluation at the end of the digital workshop 1 and they were informed before that when they filled out the evaluation, the evaluation results could be published anonymously. In the in-person presentation outside of the participatory process, we collected consent documents from each participant regarding their voluntary participation and their agreement to publish the evaluation results anonymously. In the online presentation, also outside of the participatory process, the audience was asked to evaluate the communication formats in an online evaluation and they were informed that when they filled out the evaluation, the evaluation results could be published anonymously.**"

AC: In the revised manuscript, we will adjust the reference list according to the Copernicus standards. Moreover, we will publish the script to produce Figure 10 so that scientists/experts assessing potential changes can try out the discussed uncertainty visualization formats. We will also add another uncertainty visualization format, the letter value plot, and will discuss it in comparison to the other visualization formats (see previous comments, e.g. for the methods' section introducing Table X) and make suggestions in the Discussion on how to improve the visualization formats to reduce the identified shortcomings.

---

## Author Response (AR1)

Point by point reply to comments (preprint egusphere-2023-1958 https://doi.org/10.5194/egusphere-2023-1958)

We are very thankful for the thorough reviewer comments that helped us to improve our manuscript. In particular, we changed the structure of the manuscript and added a Figure with a schematic of the presented approach and a table that presents the diverse methods for visualizing and thus communicating uncertain future climate change hazards. We modified figures and thoroughly revised the abstract, the plain language summary and the conclusions.

 Below, each reviewer's comment (indicated by "RC") is followed by our answer (indicated by "AC"). The proposed new text in the revised manuscript is written in bold. The given lines refer to the revised manuscript unless otherwise indicated.

Reply on comment of Referee #1 Usha Harris

**RC1**: Let me say at the outset that I can only comment on the participatory approach and how it may be improved, as the scientific data and models discussed in this manuscript are beyond the scope of my area of expertise.

It would be useful to have a clear definition of what the authors mean by the participatory process. Participatory processes enable ordinary people to collectively identify problems, gather information, analyse, design and identify solutions which has value to them and their network. As such an authentic participatory process includes stakeholders in all phases of the research - identification of the problem, design and dissemination of the research.

**AC:** We agree with your definition of a participatory process but do not think it is necessary to define the widely and broadly used term in our manuscript due to the following. The primary objective of the manuscript is not to provide guidance on how to conduct a complete participatory process but to concentrate on just one (important) component. We wish to offer advice on how (natural) scientists can inform stakeholders participating in a participatory process for the identification of climate change adaptation strategies about potential changes of environmental characteristics due to future climate change. This is expressed by the title of the manuscript and also the formulation of the objective in the introduction which now reads

"The objective of this paper is to show how to **quantify** climate change hazards with their uncertainties for any region around the globe from publically available ISIMIP multi-model output, and how this information can be communicated in a participatory process as a starting point for identifying local climate change adaptation strategies."

This includes guidance on how to 1) obtain quantitative information on local changes from freely available model output and 2) effectively communicate these potential changes within a participatory process. This is why we only provide the context of the communication of the climate change hazards by shortly describing the overall participatory process (formerly Section 2). In our opinion, it is sufficient to know that it was done in the first workshop, and who the participants/audience were. In addition, in Section 2.2.2.3 we described how the stakeholders further processed the provided climate change information in a World Café format. The specific knowledge generation and communication we describe may fit into different types of participatory processes.

**RC1:** A better definition for this project would be one developed by Harris (2019) specifically for environmental communication: "Participatory Environmental Communication integrates

interdisciplinary knowledge, inspires collaboration and dialogue, and utilises information networks to catalyse the agency of ordinary people towards collective action."

**AC:** Our intention was not to conduct participatory environmental communication but rather to communicate environmental data within a participatory context to empower stakeholders to make informed decisions. The participatory process extended beyond the dissemination of knowledge and continued after we communicated the knowledge presented in this article.

**RC1:** A participatory process that engages stakeholders in all aspects of the research would be difficult to fully enact in scientific research such as this which requires a high level of expertise and scientific knowledge in the field of hydrology as stated by the researchers in the following quotes:*Scientists or experts have to decide on what and how to produce climate change risk information before they communicate it to local stakeholders.*, *"An interdisciplinary team of two sociologists and us, two hydrologists, designed and conducted the participatory process." ... "The aim of all workshops was that stakeholders jointly develop climate change adaptation strategies, learn about other perspectives and network."*It is evident that the participatory process was limited to the dissemination of knowledge - *"to what effect is communicated"— to raise awareness about uncertainties and enable stakeholders to make more informed decisions in their respective roles and engage better discussions during the subsequent workshops in the participatory process of the project KlimaRhön.*

I suggest that the authors clarify:

1. How did the participants contribute (or not) to the design of the research i.e choice of study area, method, or other input into the research design?
2. What was the contribution of the sociologists in the interdisciplinary team?
3. How did the process benefit the stakeholders to make informed decisions in their respective roles?
4. Did the stakeholders jointly develop climate change adaptation strategies, learn about other perspectives and network."?

Some of these questions can be answered by conducting a focus group or distributing a qualitative research questionnaire in which the stakeholders are asked how the process benefited them *to make informed decisions in their respective roles and develop climate change adaptation strategies and learn about other perspectives and network.*"

Their comments can then be included in the manuscript. The voices of the participants would increase the credibility of the participatory process.

**AC:** Given the objective of the manuscript, we think that it is neither appropriate nor necessary to describe how the participants contributed to the research design or to describe the contribution of the sociologists as they did not contribute to how to quantify and communicate the potential changes. The quantification and communication of the potential changes was not participatory but disciplinary, but the communication was embedded in the very beginning of a participatory process. Equally, the scope of the manuscript does not encompass how the problem fields and the solutions were identified with the stakeholders and whether the stakeholders learned about other perspectives and networked in the participatory process. Consequently, we do not make assumptions about how the 30-minute communication of potential changes at the beginning of the first workshop contributed to the goals achieved in the four subsequent workshops.

However, it seems that we did not fully clarify the scope of the manuscript. In the revised version, we therefore highlighted that it involves only the quantification of the potential hydrological changes and its communication, which was set up disciplinarily – neither interdisciplinarily nor participatorily. For this, we introduced "at the very beginning of the participatory process" in line 140. And we added in lines 271-274:

"**The quantification of the potential hydrological changes and its communication method was setup disciplinarily by us, the hydrologists in the team, before the participatory process. The communication was only part of the first out of five workshops in the participatory process, which had the character of a kick-off meeting.** 31 stakeholders participated to learn about potential future changes in renewable groundwater resources and total renewable water resources by a presentation and to identify **a problem in the form of** one or two specific adaptation field(s) in a **W**orld **C**afé**.**"

**RC1:** As a communication scholar with no expertise in this area, I found it difficult to navigate the data-laden research results and terminology despite the authors' attempts to simplify the terminology and research results. Since the manuscript will be published in EGUsphere, the comprehensive engagement with data is valid. However, I would recommend less reliance on technical explanations, if they were to publish this for wider public consumption.

**AC:** In the revised version, the former Sections 3.1 and 3.2 are merged and very technical details in the newly created Section 2.1.2 are moved to the Appendix.

**RC1:** Note: This reviewer has proposed a model for participatory environmental communication which the authors may like to  consult for future projects. See

Harris, U. S. (2019). Participatory media in environmental communication: engaging communities in the periphery. (Routledge studies in environmental communication and media). London; New York: Routledge, Taylor and Francis Group.

**AC:** We sincerely appreciate your suggestions on evaluating the entire participatory process. These recommendations will be valuable when planning future articles that delve into the methodologies employed in our participatory process.

Reply on comment of anonymous Referee #2

**RC2:** General comments

The paper aims to present an approach for quantifying and communicating climate change-related uncertainties and outputs of models to stakeholders as part of a participatory climate change adaptation process. The paper provides information about the context and the participatory process used in this study, the variables used, and how results are communicated and perceived by the audience, and it discusses issues around communicating uncertainty.

The topic of the manuscript is helpful for scientists who want to identify ways to better communicate uncertainty to stakeholders, such as the uncertainty relating to climate change. The adopted approach and the study's outcomes provide insights into how communication can shape stakeholders' understanding and subsequently influence decision-making on adaptation strategies. In addition, the study provides valuable information on aspects that should be considered when designing content to visualise uncertainty.

**AC:** Thank you for the positive feedback.

**RC2:** However, the manuscript should be improved to bring out the valuable points it discusses in a more structured manner, increasing the impact of the study and broadening the readership.

First, as a general comment, the paper would benefit from some serious editing; the text is too long, and the story gets easily lost. Providing the information more concisely will make it a more valuable contribution. Specific suggestions on how the manuscript can be improved are included below.

**AC:** In the response to the specific comments, we will show where and how we made the text more concise and structured.

**RC2:** Specific comments

The paper's objective relates to two distinct tasks: (a) how to assess climate change hazards with their uncertainties from multi-model outputs and (b) how to communicate this information in a climate adaptation-focused participatory process. However, considering how the information is presented in the paper sections, it is unclear how these tasks are addressed and what the key messages are. It might help readers to follow the study more effectively if the structure of the paper was slightly re-organised. As a suggestion, the authors could distinguish the two tasks and provide details on the approach and results for each. For example, in chapter 3, where the adopted approach is presented, there could be two different clearly labelled sections, one for each objective, providing details on the methodology used to address these. Similarly, in chapter 5, the discussion could address the two distinct tasks more explicitly.

**AC:** We restructured our paper with the two foci (a) and (b) to highlight the main objectives of the paper. This is the new structure of the revised paper:

1 Introduction
2 Quantifying and communicating the uncertainty of the climate change-induced hydrological hazards
2.1 Quantification of hydrological hazard indicators
2.1.1 Study area
2.1.2 Processing of multi-model ensemble output
2.2 Communication of climate change hazards
2.2.1 Participatory process
2.2.2 Communication of the quantification
The heading of former 3.3.1 was dropped; the text is directly under the body text
2.2.2.1 Communication of potential changes of 30-year mean values by percentile boxes
2.2.2.2 Communication of potential changes in interannual variability
2.2.2.3 Summarizing hazards for the stakeholder discussion
2.2.2.4 Alternative communication of potential changes of 30-year mean values by tables
3 Results
3.1 Interpretation of communicated hazard indicators by the stakeholders (this was the last paragraph of former 3.3.4)
3.2 Evaluation of the communication format by the stakeholders of the KlimaRhön participatory process
3.3 Comparison of our communication format with a more common communication format by the audiences of two presentations of the project results
4 Discussion
4.1 Why and how should the uncertainty of hydrological changes due to climate change be quantified with a multi-model ensemble of global models?
4.2 How can the uncertainty of hydrological changes due to climate change be best communicated?
4.3 Evaluations in participatory processes
4.4 Using the uncertain information about future climate change hazards for the development of adaptation measures
5 Conclusions

Because of these changes, we also changed the short explanation of the outline of the manuscript in lines 144-146.

**RC2:** The paper includes an abstract and a plain-language summary; however, there is no significant difference between these two sections. From a typical reader's perspective and if the authors consider the need for a plain language summary, I suggest focusing on the problem this study tries to solve and why this study is important and reference the adopted approach and results using simple language to allow an average reader to understand. In addition, making reference to the usability of such an approach in a different context would also add value. In the case of the abstract, and aiming to help the readers identify the essence of the study and remember the key points, the authors could consider adding details to clarify the study's objectives and overall contribution.

**AC:** In the revised version of the abstract, we provided some more specific information and highlighted that our approach can be applied anywhere around the world due to the global coverage of the freely available data of potential future hydrological changes and also in other contexts (lines 10-14). Moreover, we structured it more clearly after the two main objectives (a) and (b) (lines 8-9; use of the term "communicate" in line 14). We have thoroughly revised the Plain Language Summary to make it more accessible to non-experts.

"Abstract. Participatory processes for identifying local climate change adaptation measures have to be performed all around the globe. **As these processes require information about context-specific climate change hazards, knowledge about how to quantify climate change hazards and how to best communicate the potential hazards with their uncertainties is essential.** In a participatory process on water-related adaptation in a biosphere reserve in Germany, we used the freely available output of a multi-model ensemble **provided by the ISIMIP initiative, which provides global coverage,** to quantify the wide range of potential future changes in (ground)water resources. **Our approach for quantifying the range of potential climate change hazards can be applied worldwide for local to regional study areas, and also for adaptation in agriculture, forestry, fisheries, and biodiversity**. To support participatory climate change adaptation processes, we propose to **communicate** uncertain local climate change hazards with percentile boxes rather than with boxplots or simple averages **with the** model agreement on the sign of change. This **helps the** stakeholders **in** identifying the future changes they wish to adapt to depending on the problem (e.g., resource scarcity vs. resource excess) and their risk aversion. Using or adapting our **quantification** and communication approach, flexible climate change risk management strategies can and should be developed worldwide in a participatory and transdisciplinary manner involving stakeholders and scientists."

"Plain Language Summary. **All around the world, it is necessary to adapt to climate change, and people need to work together in local participatory processes to be able t**o identify **the best local adaptation measures**. **Any development of adaptation measures requires information about the changes that may occur** in the future**, for example changes in water resources or crop yield**. **As the future cannot be reliably predicted, a range of possible future changes should be considered. These can be quantified with** free data **of global coverage** from multiple computer models**, which is available for many sectors like water, agriculture, forestry, fisheries, and biodiversity**. To **optimize communication**, we propose using "percentile boxes" instead of boxplots or simple averages **with the** model agreement on whether there will be an increase or a decrease in water resources. This way, people can better understand **what may happen in the future** and decide what possible future they want to adapt to, **for example** to much less or to somewhat less water than today, depending on how much risk they are willing to take. Our **quantification** and communication approach can support climate change adaptation processes worldwide, where stakeholders and scientists collaborate to develop flexible strategies for reducing climate change risks.

**RC2:** The introduction section seems too long and makes it difficult for the average reader to understand the background, context, and problem this study aims to solve. Also, how it is written makes it difficult for the reader to identify the gap in the literature this paper addresses. The suggestions below can improve the contextualisation of the study and will help the reader follow the next sections better:

- **RC2:** Provide information on existing approaches; this could be a table presenting advantages and disadvantages and highlighting challenges when communicating uncertainties. The proposed table will allow the reader to understand the existing approaches, what is missing, and what this paper seeks to address.

  **AC:** In the discussion (section 4.2), we introduced Table 1 as Table 1 explaining the advantages and disadvantages of uncertainty visualization to communicate uncertainty to stakeholders. Due to the introduction of Table 1, we deleted some text parts in Section 4.2. Moreover, we explained why we did not use existing uncertainty visualizations but used an own visualization format, which is a modified visualization of the GERICS bar chart (lines 545-554) in the discussion. We presented existing uncertainty visualization formats in the introduction in lines 94-113 and opened the gap that a suitable uncertainty visualization format for climate change adaptation processes needs to be identified (lines 113-114).

**Table 1. Advantages and disadvantages of some visualization methods to communicate uncertain changes to stakeholders. All represent the distribution of changes that are simulated by the different members of the multi-model ensemble and indicate the change values that are not exceeded by a certain percentage (percentile) of the ensemble members.**

| Uncertainty visualization | Advantages | Disadvantages |
|---|---|---|
| Boxplot | Shows three percentiles (P25, P50, P75) and possibly outliers and minimum (P0) and maximum (P100) values; common visualization. | Potentially reduced readability if it shows minimum and maximum; non-unique definition of and difficult to interpret "whiskers"; does not show percentiles suitable for strongly risk-averse stakeholders (P10 and P90); does not show the distribution of simulated change values precisely. |
| GERICS bar chart | Shows five percentiles (P0, P20, P50, P80, and P100). | Potentially reduced readability as it shows minimum and maximum; does not show the distribution of simulated change values precisely. |
| Violin plot | Precisely visualizes uncertainty by displaying the distribution of simulated change values within the whole range by the width of the box. | Potentially reduced readability as it shows minimum and maximum; only shows the percentiles minimum, maximum, and median (does not show many percentiles); smoothing of the shape leads to only an approximate representation of the distribution; the high information content due to the high uncertainty precision might be overwhelming and difficult to read for the end user. |
| Letter-value plot | Shows many percentiles and uses the width of the boxes to guide the eye. | Potentially reduced readability as it shows minimum and maximum; area of the boxes does not correspond to the number of simulated change values contained in the boxes; pseudo-(i.e., misleading) visualization of the distribution of simulated change values; depending on the visualization, it has too many percentiles. |
| Percentile box | Shows five percentiles; percentiles can be selected depending on the problem and the risk aversion; minimum and maximum can be avoided, which increases the readability of the more central values and prevents the misinterpretation that the minimum and maximum values of the ensemble are the most extreme values possible. | Does not show the distribution of simulated change values precisely. |

- **RC2:** There is information in other sections of the manuscript that can be used in the Introduction section to improve the contextualisation of the study. For example, in section 5 (§5.1.1 and 5.1.2), information around uncertainty and available models and visualisation formats could be used to set the background, preparing the reader about what is coming up and what this study is trying to achieve. Similarly, information included in section 6 around the theoretical background (1ˢᵗ paragraph on page 29 - lines 741-752) could be used to shape the contextualisation of the study.

  **AC:** We structured the introduction in a way that better highlights the main objectives (a) and (b). For this, we introduced one sentence, which outlines the structure of the introduction in lines 52-53 "**This is why (a) future changes should be assessed with their uncertainty and (b) a suitable visualization should be found with which the future changes with their uncertainty can be communicated.**"

  In line 94, we added "**To communicate the quantified potential changes with their uncertainties visually, a suitable visualization format is needed and** should not be a translation into median or mean changes only, […].".

  Then, in the following paragraph, we included information from the former 5.1.1 and 5.1.2. For this, we moved a paragraph from the Discussion to lines 69-74 in the Introduction.

  In line 111-113, we introduced:

  "**Similarly, the letter value plot shows several percentiles with bars, but with reduced bar width the more distant it is to the median. To show the distribution of values, violin plots can be used, which also show the minimum, median, and maximum values.**"

  As we have not embraced the Cultural Theory in the communication of the uncertainties in the participatory process, we did not move lines 709-720 (formerly lines 741-752) from the conclusion to the introduction but leave it in the conclusion as a recommendation.

  To highlight the two main objectives, we changed synonyms (e.g. "assess" or "present") to the two words "communicate / communication" and "quantify / quantification (e.g., line 132, 138).

- **RC2:** To reduce the length of this section and improve its readability, I suggest making the presented information more concise and reducing repetition. Removing the 2ⁿᵈ paragraph on page 3 (lines 76-89) and adding the content as supplementary material would help reduce the length and allow the typical reader to focus on the context of the study.

  **AC:** We find the paragraph on ISIMIP very valuable for persons who seek to analyze data as we did for another sector to motivate that our approach is also possible with other than hydrological data. But, in the revised version, we shortened the paragraph in lines 75-81 as shown:

  "**The Inter-Sectoral Impact Model Intercomparison Project (ISIMIP, www.isimip.org) provides freely available multi-model ensembles of many model output variables that are of interest to quantify climate change hazards in several impact sectors (water, lakes, biomes, forests, permafrost, agriculture (crop modelling), energy, health, coastal systems, fisheries and marine ecosystems, and terrestrial biodiversity; ISIMIP, 2019). The available impact model outputs mostly cover all land areas of the globe. For each impact variable, ISIMIP2b provides a time series for historic and future periods and several greenhouse gas emissions scenarios, which were computed by multiple global impact models (Frieler et al., 2017), with which the uncertainties of future changes in impacted variables can be characterized.**"

  The rest of the paragraph was moved to Appendix A4. Moreover, we moved former lines 144-148 to Section 2 (lines 206-210) and moved the explanation of impact models in former lines 52-59 to Appendix A2 and only shortly introduced the topic in lines 54-55.

**RC2:** Section 2 provides information about the case study and the participatory process. The authors could consider if this section could be incorporated into the methodology and results chapters. For example, the case study part §2.1 presents the study area and, therefore, could be part of section 4,

which presents the study results. Also, the participatory process part §2.2 discusses the approach adopted in this study and links to the methodological aspects. As a suggestion, this part can be included in Section 3, which discusses the overall methodology. In this way, the information will be presented in a more structured way, allowing an average reader to follow the paper better.

**AC:** In the revised version, we moved both Sections in Section 2 (as Sections 2.1.1 and 2.2.1).

**RC2:** Section 3 presents the approach adopted in the study. This section seems too long and makes it difficult for an average reader to understand the approach and reasoning behind it. I suggest the following changes that will improve the way the information is captured and will enable the reader to understand easily the adopted process, broadening the readership of the manuscript:

- **RC2:** The authors could explain the methodology used per objectives (a and b) as mentioned previously; there could be two different clearly labelled sections, one for each objective, providing details on the methodology used to address these. Also, to allow readers to follow the logic of the methodology more closely, the authors could consider including a graphical abstract of the main steps of their approach per objective.
  **AC:** As shown previously, we changed the structure of this Section (Section 2). A schematic of the presented approach was included as Figure 1 at the beginning of Section 2 and now guides the reader early on.

[Figure]

**Figure 1. Schematic of the presented approach of quantifying and communicating uncertain climate change hazards in participatory climate change adaptation processes. ISIMIP: The Inter-Sectoral Impact Model Intercomparison Project (www.isimip.org/).**

- **RC2:** Sections 3.1 and 3.2 provide information on the hydrological hazard indicators; although the information is useful, it can distract the reader from understanding the adopted approach. I suggest the authors provide a brief summary in this section and include the information as supplementary material to improve the readability.
  **AC:** In the revised version, the former sections 3.1 and 3.2 are merged (see above). The text is not structured by subsubsections anymore and was made more concise (very technical details important to applying the approach are moved to the Appendix Sections A3, A5, A6, A7, A8, and A9). To make it more concise, we only left the first two sentences of the paragraph "Greenhouse gas emissions scenarios" (the former 3.1.2) in the text, the rest was moved to Appendix A6.
- **RC2:** Sections 3.1 and 3.2 provide information on the analysis conducted, addressing the objective (a) of the study; what is missing to allow readers to follow the approach more closely is the output of this task, what this analysis provides and how it feeds into the next step (objective b). Currently, this information is captured in section 3.3; for example, the analysis output was the design of graphics used in the participatory process (as mentioned in §3.3.2, line 325).
  **AC:** To make the connection between the objectives (a) and (b) and due to the inclusion of a

graphical abstract, we added a first sentence and slightly rewrote the first paragraph in Section 2 (formerly Section 3):

"**Future changes should be quantified with their uncertainty and then a suitable visualization should be found with which the future changes with their uncertainty can be communicated in participatory climate change adaptation processes; an approach that we applied (Figure 1). At first, s**cientists or experts have to decide on what and how to produce climate change risk information **(Figure 1, left box), and how to visualize the information (Figure 1, arrow between boxes)** before they communicate it to local stakeholders **(Figure 1, right box)**. So, during the first step, the **quantification,** they need to decide what indicators of climate change hazard should be quantified, given the problem, the interest of the stakeholders, data availability and quality as well as technical and time constraints. **In the second step,** the scientists, experts or communicators have to decide on what, **with which visualization format** and how to communicate given their audience, the aim and the generally severe time constraints in the participatory process."

Moreover, we included a graphical abstract, in which the two steps quantification and communication are displayed, which are connected by an arrow, which says "decide for uncertainty visualization for each indicator".

- **RC2:** Section 3.3 provides information on communicating the hazard indicators, so it mainly addresses the objective (b) of the study. Although the section provides detailed information on the process followed, how the information is presented can be confusing for an average reader. The authors could consider (as mentioned previously) including the participatory process part §2.2 in this section, as it addresses methodological aspects adopted in this study. The overall process and information could be presented in a graph or a table, showing details on the workshops, timeline, participants, objectives, and what has been achieved. Also, the authors could consider adding information to reflect why they chose the specific way of communication to allow the reader to understand the driver behind the specific approach. Furthermore, information relevant to the options provided to stakeholders (mentioned in section 4, page 21) could also be included here. This will improve the paper's readability, allowing the reader to follow the logic of the participatory process more closely.

  **AC:** As mentioned above and proposed by the referee, we moved the Section about the participatory process to the communication section (Section 2.2.1). Moreover, we moved lines 399-405 (formerly lines 505-511 in Section 4.2) and Figure 7 (formerly Figure 8) to Section 2.2.2.4. To help the reader understand our communication approach (in vs. outside the participatory process), we included lines 290-297, and made some small changes in the text (e.g. in the caption of Figure 7 or e.g. including the words "participatory process" or "outside of the participatory process" in the text where needed).

  To explain why we chose the specific visualization, we included lines 545-554 and the table of the advantages and disadvantages of uncertainty visualizations (Table 1) in the Discussion (Section 4.2). However, we think that more information on the participatory process will not enhance understanding as the paper only covers the very first workshop in the participatory process.

- **RC2:** Section 3.3 is too long and includes details that can confuse the average reader. To improve the readability, I suggest the authors consider including information that is relevant to the approach only and not the results and presenting it more concisely. For example, the last paragraph of the section on page 18 (lines 442-454) presents some of the results that can be included in section 4 (results).

  **AC:** In the revised version, we introduced Section 2.2.2.3 for the former 3.3.4 and reduced it by the second paragraph, which we moved to Section 3.1 (i.e. to the results as suggested).

- **RC2:** The authors could consider adding a more detailed description in Figure 2 (page 13) to allow the reader to understand why the selected multi-model ensemble represents the currently best estimate of future hydrological hazards.

  **AC:** Figure 2 shows exactly the slide that we showed and explained to the stakeholders. We wanted to show the reader exactly what we showed the stakeholders, and therefore do not want to add anything to Figure 3 (formerly Figure 2). Later, in Section 4, we discuss the advantages and disadvantages of the chosen multi-model ensemble (lines 494-508; 521-527).

- **RC2:** Average readers might be able to follow the graphs presented in Figure 4 (page 15) more clearly if there was a more thorough explanation of the potential change of groundwater recharge in the caption, as mentioned on page 14 (lines 345-352). A similar approach could be adopted for Figure 5).
  **AC:** It is not the main intention of the article to present our interpretation of the processed potential changes but to show how we communicated the potential changes and how we communicated our interpretation of the potential changes. Therefore, we find it too exhaustive to add the interpretation in the figure caption. But we now refer to lines 324-331 in the caption of Figure 5 and lines 364-370 in the caption of Figure 6 in the revised version.

- **RC2:** In section 3.3, pages 14-15, lines 359-368, information more relevant to the analysis rather than the communication aspects is included. The authors could consider whether this information can be moved to section 3.1 (as per the previous comment) to allow readers to follow the logic of the methodology more closely.
  **AC:** In the mentioned paragraph, the processed data is interpreted (also in the paragraph in the former lines 405-416), which is neither part of our communication approach nor part of our quantification approach. We moved these two parts in the Appendix Sections A10 and A11 and refer to them in the captions of Figures 5 and 6 respectively, which is not the perfect solution because the interpretation should be close to the visualization of the results for better comprehension.

- **RC2:** To reduce the length of this section, terms and concepts (including Table 1) could be included in the introduction or supplementary material.
  **AC:** In the revised version, we moved the former Table 1 into the Appendix A1 as Table A1.

**RC2:** Section 4 presents the results of the study. As mentioned in a previous comment, this section could include information about the case study (currently §2.1) at the beginning and then present the results.

**AC:** In the revised version, we moved the former Section 2.1 before Section 2.1.2.

**RC2:** Figures 6 and 7 could be presented on the same page, allowing the average reader to compare the results easily.

**AC:** To make sure that former Figures 6 and 7 are presented on the same page, we rearranged them and displayed them in one figure, Figure 8 (see below). Because of this change, we had to make some changes in the text in lines 438-440 and in the caption of Figure 8.

[Figure]

**RC2:** The authors could consider adding information on the importance of the results and how these are dependent on the number of participants in the workshops. For example, based on the online participation, the results seem more significant than those from the in-person presentation. There is no information to determine whether the type of participants in the in-person workshop adds value to the results. Also, no information is included regarding the conducted statistical analysis of the results.

**AC:** The results of the evaluation by the stakeholders in the participatory processes is more important than those of the in-person presentations as we wanted to study methods for participatory processes on climate change adaptation. This is expressed in lines 610-611, 648-650 and in Section 4.3 in general. As we wanted to address all types of persons with our uncertainty communication, we did not make differences between the types of participants. We discuss in Section 4 that the information needs and the intention of listening to the uncertainty communication (interest in the presentation of project results vs. participation in a workshop to develop adaptation measures) is important for the suitability of uncertainty communication. Due to the low number of respondents, no further statistical analysis is deemed to be necessary.

**RC2:** Finally, the authors could consider a different type of format for Figure 9 (perhaps combining the two visualisations in one) to avoid confusion with the graphs used in the participatory process and help the reader interpret the results. In addition, and as per the GC guidelines, the authors should use in the figures a colour combination that would allow readers with colour vision deficiencies to interpret the findings correctly.

**AC:** To avoid confusion, we extended the figure captions of Figures 8 and 9 to clarify that these evaluations refer to two presentations of the project results to persons that have not been involved in the participatory process. We also added in the titles of Figure 9 and the combined Figure of Figures 6 and 7: "Evaluation by audiences of presentation of project results" and to the combined Figure also: "Evaluation by participants of the climate change adaptation process" (see Figures below and above).

In the revised version, we also changed the color combination for better visual readability (Figure 9, which is the figure below). The color combination is colorblind safe and print friendly concerning https://colorbrewer2.org/.

[Figure]

**RC2:** The discussion chapter (Section 5) could be further improved by adding more detail on interpreting the results in the context of the objective of this study rather than referencing the results, which is already done in Section 4. For example, considering the outcome of the workshops, did the authors further explore the stakeholders' views on the communication format that would make more sense to them?

**AC:** Within the scope of this paper, we did not explore stakeholders' views in addition to the evaluations that we presented in this paper. The aim of the whole participatory process (in which this presented study just represented an aspect in the first of five workshops) was to identify adaptation measures in water management for the study area. We aimed to ensure that stakeholders did not perceive themselves as experimental subjects and recognized that their identification of measures was taken seriously, not merely as an observation of communication methods.

**RC2:** How did the authors interpret the results considering the overall context of the study?

**AC:** In the revised version, we included in lines 656-659 that the specific results of the multi-model ensemble were not used quantitatively (only qualitatively) in the discussion of adaptation measures because no technical measures were discussed or monetary cost-benefit analyses were performed.

**RC2:** What are the key messages?

**AC:** We included the following information in the conclusion (Section 5):

Concerning objective (a):

- To assess uncertainties with multi-model ensembles it is more robust using changes (not absolute values) (lines 693-694).
- Moreover, our approach is preferable to use even for local study areas except when another (local) multi-model ensemble is available, which is very rarely the case (lines 688-693).
- The important changes of interannual variability were hard to grasp for non-scientists, thus, for their communication, another communication format is needed (lines 707-708).

Concerning objective (b):

- Based on our experience and the results, we believe that for stakeholders who need to identify climate change adaptation measures hazard communication by percentile boxes is preferable to communication by simple tables (lines 695-700).
- For the presentation of climate change hazards to the general public, a simple table with the mean changes and an indication of the agreement of the models on the sign of change is preferable (lines 701-703).

**RC2:** What is the novelty of the approach, what are the advantages and disadvantages, and why was this specific approach chosen? What are the limitations?

**AC:** The novelty of our approach is that we highlighted that multi-model ensembles exist and that their results are globally available (e.g. line 139). We show how to make use of a global multi-model ensemble analysis for a small study area in very detail (processing can be done with basic knowledge in any programming language such as R, Python, or MatLab) so that it can also be done worldwide. This is especially useful for locations where no local multi-model ensemble is available. Moreover, in the paper, we show in very detail how uncertainty can be communicated with various uncertainty visualization formats in a real participatory process. We combined these novelty aspects with the key messages and revised the Discussion and the Conclusions with the information (see previous answer and below).

In the revised version of the Discussion, we included in Table 1 (last row, second column):

"**Shows five percentiles; percentiles can be selected depending on the problem and the risk aversion; minimum and maximum can be avoided, which increases the readability of the more central values and prevents the misinterpretation that the minimum and maximum values of the ensemble are the most extreme values possible.**"

To stress the disadvantages, we included in lines 577-578:

"**However, the communication, i.e. the explanation of the approach and the interpretation, takes a lot of time in a workshop and asks the stakeholders for a long concentration span.**"

Moreover, in lines 529-554, lines 570-584 and in Table 1, we showed the advantages and disadvantages of our uncertainty visualization formats compared to other uncertainty visualization

formats. And in lines 555-565, we included ideas how other uncertainty visualization formats could be improved for the communication in participatory climate change adaptation processes.

At the end of the first paragraph (line 507-508), we included a technical limitation:

"**However, it needs an expert with basic technical knowledge in any programming language to quantify the potential changes.**"

In lines 545-551, we explained why we chose our approach:

"**We came up with our approach because we wanted to communicate more percentiles and did not want to communicate minimum and maximum values. We left out the minimum and maximum 1) to not give too much room to possible outliers and 2) to not give the impression that minimum and maximum values could not be exceeded in reality. We wanted to have several percentiles to have the possibility to say how many percent of models simulated a stronger or weaker change than a certain change value, which supports communication that does not include a specific prediction. Another advantage is that the percentiles can be chosen individually depending on the risk aversion and the problem (Table 1).**"

In lines 688-690, we included:

"**The presented method for producing quantitative estimates of future climate change hazards, which benefits from the freely available output of global multi-model ensembles (provided by the ISIMIP initiative), can be replicated by anybody with basic knowledge in any programming language such as R, Python, or MatLab.**"

**RC2:** Also, a typical reader might not understand how the specific approach adopted in this study tackled some of the issues mentioned in the chapter. For example, how did it address the transparency?

**AC:** In lines 619-621, we included that we showed the stakeholders the uncertainty of the models through multi-model ensemble and showed them the approach of our quantification to address the uncertainty routine "transparency".

"**By showing the stakeholders the uncertainty of the models through multi-model ensembles and showing them the approach of our quantification,** we address one of the uncertainty routines of stakeholders called "transparency", in which the stakeholder **considers** the limits of knowledge (Höllermann and Evers, 2019)."

**RC2:** How did it help the stakeholders in decision-making? Explicitly exploring and answering these questions would lead to a much stronger paper with a more broadly applicable impact and will allow for broadening the readership of the manuscript beyond subject area experts.

**AC:** We cannot make reliable statements about how our approach helped the stakeholders in decision-making because we guess (and hope) that the whole participatory process influenced their decision-making and because we did not evaluate how their decision-making changed. We will integrate in the Discussion that we hope to have highlighted uncertainty enough so that they will more carefully look at uncertainties in their decision-making in the future. In line 666-667, we included:

"**Due to highlighting the uncertainty of future changes, we hope that the stakeholders will more carefully embrace uncertainty in their decision-making in the future.** Next to the uncertainty of…"

**RC2:** The conclusions section (Section 6) would benefit from a synthesis of the main points of the study, highlighting the advantages of the adopted approach and the importance of the paper more

briefly. This would help the reader understand why the study should matter to them after having finished reading the paper.

**AC:** We completely rewrote the Conclusions in the revised version integrating the key messages, novelty aspects, and advantages formulated above and structuring after the objectives (a) and (b).

**RC2:** Finally, as a suggestion and according to the GC guidance, the ethical statement should be more comprehensive, and a description of the process should be included in the methodology section of the paper.

**AC:** We added following sentences in lines 841-847 to make the ethical statement more comprehensive:

"**For this, the stakeholders were asked to evaluate (among others) the communication format in an online evaluation at the end of the digital workshop 1 and they were informed before that the evaluation results may be published anonymously. In the in-person presentation outside of the participatory process, consent documents were collected from each participant regarding their voluntary participation and their agreement to publish the evaluation results anonymously. In the online presentation outside of the participatory process, the audience was asked to evaluate the communication formats in an online evaluation and they were informed that the evaluation results may be published anonymously.**"

Other changes

**AC:** In the revised manuscript, we adjusted the reference list according to the Copernicus standards and according to the changes made in the revised manuscript.

We published the script to produce Figure 10 so that scientists/experts assessing potential changes can try out the discussed uncertainty visualization formats (see https://zenodo.org/records/10400312). We added this information in lines 542-543, and 825-826.

We added another uncertainty visualization format, the letter value plot, discussed it in comparison to the other visualization formats (see previous comments, e.g. in introduction of Table 1), and made suggestions in the Discussion (Section 4) on how to improve the visualization formats to reduce the identified shortcomings (lines 556-565).

We moved the text in the former lines 312-314 to lines 249-251 to enhance understanding.

We did some spelling corrections.

---

## Author Response (AR2)

Point by point reply to comments (preprint egusphere-2023-1958 https://doi.org/10.5194/egusphere-2023-1958)

We are very thankful for the thorough reviewer and editor comments that helped us to improve our manuscript. In particular, we moved parts of the Discussion that presented the diverse methods for visualizing and thus communicating uncertain future climate change hazards to the beginning of the manuscript, and reduced the length. We modified figures and revised the abstract and the conclusions.

Below, each reviewer's comment (indicated by "RC") and editor's comment (indicated by "EC") is followed by our answer (indicated by "AC"). The proposed new text in the revised manuscript is written in bold. The given lines and sections numbers refer to the revised manuscript unless otherwise indicated.

**Reply to comment of anonymous Referee**

**RC:** General comments

The authors have addressed the comments and suggestions made as part of the initial review and explained the reasoning for the amendments they made to the manuscript, which now brings out the valuable points it discusses more effectively, increasing the study's impact.

**AC:** Thank you for the positive feedback.

**RC:** Specific comments

The introduction of Table 1 to the manuscript provides readers with an overview of the advantages and disadvantages of some visualisation methods to communicate uncertain changes to stakeholders. To further improve the manuscript's readability, the authors could consider moving Table 1 to the Introduction instead of the Discussion section. Presenting this information at the beginning of the paper would enhance the background and conceptualisation of the study and will make it easier for the reader to focus on the topic and what the study aims to achieve.

**AC:** We introduced a new Section 2 that provides the table and figure with the overview of visualization methods to communicate uncertain impacts of climate change (formerly in the Discussion section).

**RC:** Finally, there is a spelling error in Figure 8, which the authors can correct.

**AC:** In Figure 8 and its caption, we changed the statement to "Through the scientific input / the results (figures) just presented, I gained a better understanding of the potential changes **of** water resources due to climate change in the biosphere reserve Rhön".

**Reply to comment of Editor**

**EC:** The authors have addressed many of the reviewers points, and the manuscript benefited from the suggested restructuring so far (e.g., to clarify the 2 different foci, as suggested by reviewer 2).

**AC:** Thank you for the positive feedback.

**EC:** However, there are a few additional points that I believe need to be addressed prior to the manuscript's publication. Most of them pertain to improving the manuscript's clarity. I am leaving some general and specific suggestions below. Some of these incorporate or build on the reviewers' recent comments, others are based more around GC standards.

General comment and suggestion:

GC1. It can be tricky in parts of the manuscript to figure out exactly what the "experiment" was and how the results lead to certain conclusions. Think of the manuscript as a classic research paper centred around a "knowledge finding process" or experiment. This will help you to keep the manuscript more focused and identify parts of the text that may not be needed here. Overall, the manuscript is longer than necessary, and it compromises clarity.

**AC:** The objective of our manuscript is to support other scientists in participatory climate change adaptation processes. We never thought of our study as an experiment in which we wanted to "prove" that a hypothesis is true (or similar). If we would talk about an experiment, then Section 3.1 describes the base of the experiment and Section 3.2 describes the experiment.

We have tried to see our manuscript more like the description of a knowledge finding process, as recommended by you and with this in mind make it more concise. As a consequence, we added two sentences to the abstract (see our response to SC1), the information about the study area was deleted and is now provided in the Supplements for interested readers. The text in line 236 was changed to "[…] selected the data only of the **four** grid cells […]". Moreover, we deleted some sentences and formulated a few others more concisely.

**EC:** GC2. Make sure you provide information on existing approaches in the introduction (or methods when strongly method related) when it is unrelated to/not based on your own work presented in the paper. Both reviewers commented on this in some form, and I strongly agree.

**AC:** To provide information on approaches using ISIMIP multi-model ensembles, we refer to two additional studies (Lange et al., 2020 and Tabari et al., 2021) in the Introduction (lines 57 and 60). Moreover, we refer to the original studies that introduced the uncertainty visualization formats in the Introduction (lines 104-115).

To provide more information on problems in evaluations in participatory processes, we added two references including two sentences in the Discussion in Section 5.3:

Lines 610-612: "**In participatory processes, no controlled experiments are possible due to the nature of participatory processes and ethical reasons (Lange et al., 2021).**"

Line 619-620: "**Even in participatory adaptation processes, the participants'expectations and needs for information differ (Rosener, 1981). In general, we can distinguish** […]"

**EC:** Specific (minor and major) comments and suggestions:
* * *
SC1. Abstract: Make sure to highlight the research aspect of the work in the abstract, i.e. clarify where your conclusions come from. The abstract should be clear in what was the "experiment" or "knowledge finding process" in a broader sense, and what knowledge was gained from it. You have gathered information (presented in your results section here), and it should be clear that your conclusions are based on it. The experiments (incl. Mention of the questionnaires) and results should be highlighted briefly before your conclusions.

**AC:** In lines 14-21, we added/rewrote the following sentences: "**We evaluated our approach to communicating uncertain local climate change hazards by questionnaires for the stakeholders in the participatory process and the audiences of two project results presentations for the general public. To support the stakeholders in participatory climate change adaptation processes,** we propose to **use** percentile boxes rather than **boxplots for visualizing the range of potential future changes**. This helps the stakeholders identify the future changes they wish to adapt to, depending on the problem (e.g., resource scarcity vs. resource excess) and their risk aversion. **The general public is best informed by simple ensemble averages of potential future changes together with the model agreement on the sign of change.** Using or adapting our quantification and communication approach, flexible climate change **adaptation** strategies can and should be developed worldwide in a participatory and transdisciplinary manner, involving stakeholders and scientists."

**EC:** SC2. Remove all line references from the manuscript, incl. figure captions (fig. 5) - lines will change and not be numbered in the final manuscript. refer to (sub)sections instead.

**AC:** We removed the line references in the captions of Figures 5 and 6 and inserted instead the reference to the subsection.

**EC:** SC3. L39-40: Grammar is a bit off around "e.g." and difficult to read. Consider rephrasing the sentence, omitting the example or moving it to another sentence for clarity.

**AC:** We omitted the example.

**EC:** SC4. L161: Correct to "[…] who have diverse [...]" or "[...] who come from diverse [...]"

**AC:** We corrected to "[…] who have diverse [...]".

**EC:** SC5. The most recent review points to a spelling mistake in Figure 8. Please find and correct it.

**AC:** In Figure 8 and its caption, we changed the statement to "Through the scientific input / the results (figures) just presented, I gained a better understanding of the potential changes **of** water resources due to climate change in the biosphere reserve Rhön".

**EC:** SC6. Section 4.2: Much of section 4.2 provides background and a general discussion of visualisations, and an overview table (table 1) that would be useful earlier in the manuscript – I agree strongly with the reviewers here. I suggest moving it to the beginning of section 2 or even to the introduction (in a subsection providing some background to communication/visualisation). Keep in mind that the discussion section primarily serves to discuss your results and put them in context of previous work.

**AC:** We introduced a new Section 2 that provides the table and figure with the overview of visualization methods to communicate uncertain impacts of climate change (formerly in the Discussion section).

**EC:** SC7. Conclusions: Make sure it is clear to your reader where your conclusions come from, i.e. how can you make these statements, what evidence did you gather for it in your study (even if that evidence is only weak). Be specific in this; Rather than writing "Based on our experience" (L695), let

the reader know what that experience (and evidence) is. E.g. "Based on the (questionnaire-based and informal) responses of participants, we can conclude ..."

**AC:** In lines 678-680, we rewrote the sentence as suggested:
"Based on **the questionnaire-based evaluations of the participants,** we **can conclude that different** formats **for communicating** the range of potential future changes **should be used when addressing** either **the** stakeholders in a climate change adaptation process or the general public."

**EC:** SC8. Make sure to comment on the statistics or poor suitability for specific statistical analyses (RC2). If your sample numbers are too low, it would help if you highlighted the statistical tests you would/could conduct with different numbers. This will help others build on your study, for example. at the same time, you acknowledge important limitations here. This can be a small part of the discussion.

**AC:** In participatory processes, it is not (or even never) feasible to invite a significant number of participants to retrieve a significant evaluation that would enable a statistical test. Similarly, it is not practical to establish a control group with alternative treatments, as is common in clinical studies, due to the associated organizational and time constraints. With the manuscript, we aim to present and discuss experiences of the communication of uncertainty in a participatory process to support others in designing a participatory process. Even if statistical tests could be conducted, the various natures of participatory processes (due to the problem field, involved stakeholders, involved scientists, country, etc.) would make the statistical test results useless.

To highlight the limits of the evaluation, we included in lines 135-136 "**While the communication approach was evaluated, no general conclusions can be drawn due to the small number of evaluating participants.**".

**EC:** SC9. Fig. 8: Adopt fig. 9 colours (to make figures consistent and all suitable for types of colour blindness). The font size (for legends in both figures and longer text blocks in fig. 8) should also be increased to make them more accessible.

**AC:** Figures 8 and 9 were re-designed to adjust text size and to adjust the colors to make it suitable for many types of color blindness (according to Crameri, 2018). We did not align the colors in Figures 8 and 9 because in Figure 8, we need a diverging color palette for the discrete data type, while we need a sequential color palette for the categorical data in Figure 9. However, we chose the two color palettes from Crameri (2018) suitable for our data and resembling the most.

Other changes

**AC:** We corrected spelling errors. However, we did not follow the remarks from the precedent review file validation because Figure 6 was presented in this way to the stakeholders in the first workshop. But we discuss in lines 563-564 that the colors in Figure 6 should be changed to colors from color palettes visible for persons with color vision deficiency.

---

## Author Response (AR3)

Point by point reply to comments (preprint egusphere-2023-1958 https://doi.org/10.5194/egusphere-2023-1958)

We are very thankful for the executive editor's comments that helped us to improve our manuscript. Below, each executive editor's comment (indicated by "EC") is followed by our answer (indicated by "AC"). The proposed new text in the revised manuscript is written in bold. The given lines and section numbers refer to the revised manuscript unless otherwise indicated.

**EC:** Thank you for your revisions and meaningful engagement with the review process. We are happy to publish this work after a few more minor revisions. Please read my suggestions below for more details.

**AC:** Thank you for the positive feedback.

**EC:** In general, the abstract can be re-structured to better communicate what the authors have done and the findings. For example, some sentences from the introduction section can be adapted and used in the abstract (e.g., in this study, we show how to quantify climate change hazards with their uncertainties for any region around the globe from publicly available ISIMIP multi-model output. We also demonstrate how climate change hazard uncertainties can be communicated effectively for identifying local climate change adaptation strategies.). Then add 2-3 sentences how this was done and also list the key findings.

**AC:** To follow your advice, we have reformulated the second sentence of the abstract, which now reads:
"As these processes require information about context-specific climate change hazards, **we show in this study** how to quantify climate change hazards **with their uncertainties in regions all around the globe** and how to best communicate the potential hazards with their uncertainties **for identifying local climate change adaptation strategies.**"

How this was done and the key findings have already been provided in previous of the abstract.

**EC:** Lines 14-15 (in the abstract) - It is not clear what the authors are stating especially the last part of the sentence. Please consider revising this sentence (e.g., We evaluated our approach to communicating uncertain local climate change hazards by questionnaires for the stakeholders in the participatory process and the audiences of two project results presentations for the general public').

**AC:** In lines 14-16, we changed the sentence as follows:

"We evaluated our approach to communicating uncertain local climate change hazards by questionnaires **that** the stakeholders in the participatory process and the audiences **from the general public** of two project results presentations **answered**."

**EC:** Sometimes the authors refer to themselves as 'we' and 'the authors of this paper'. Please pick one and use consistently.

**AC:** We checked the text and now use "we" consistently. Therefore, in line 141, we now use "we" instead of "the authors of this paper".

**EC:** Line 135: "While the communication approach was evaluated, no general conclusions can be drawn due to the small number of evaluating participants." Insert in parenthesis, the number of those evaluated. The number ranges from 7 to >40 so it is not clear what exactly the authors are referring to. Also, consider replacing 'concrete' with 'general' in this sentence.

**AC:** In lines 137-138, we changed the word "general" to "concrete" and added "in a participatory process (26 evaluating participants)" at the end of the sentence.

**EC:** Please include the questionnaire used with the stakeholders as Supplemental Materials. The authors also mention there was an interview session with 22 stakeholders before the start of workshop 1. Was there a questionnaire for these interviews too? Can the findings be included in the text or as Supplemental Materials?

**AC:** In the supplement, we added S.2 with the questionnaire for the evaluation of our communication format with a more common communication format and added "(the questionnaire can be found in Supplement S.2)" in the manuscript in line 474. The questionnaire questions and the answers of the stakeholders in the first workshop of the participatory process can be found in Müller and Czymai (2022), which is referenced in the manuscript. The questionnaire for the interviews and the results had nothing to do with the quantification and visualization of potential climate change hazards with their uncertainty, which is why we decided not to include it in the text or the supplemental materials.

**EC:** Please consider revising sentences where the term 'weak evaluation' is used, and instead explain the limits of your evaluation strategies. Weak is a subjective word and can be interpreted in different ways.

**AC:** In lines 609-610, we added "because of the small number of evaluating participants and the high context dependence" to explain why we consider the evaluation to be weak.

**EC:** In Table 1, list visualization methods in the same order as Figure 1 (left to right), or even better (if possible) consider combining figure 1 and table 1 so the reader doesn't have to go back and forth between these two charts.

**AC:** Thank you very much for your suggestion. We rearranged and integrated the former Table 1 into Figure 1.

**EC:** Remove informal writing (example: line 195 'So, ...')

**AC:** In line 193, we replaced "So" with "Thus". Moreover, we checked the whole text to remove informal writing (e.g. "even more so" was replaced by "particularly" in line 59, "just like" was replaced by "with the same approach as" in line 312).

**EC:** I suggest removing Figure 2 as it is explained clearly in the text. However, if you want to keep it, consider making some revisions to increase its readability. For instance, label each step on the figure (Step 1, Step 2, etc.). Also, consider adding figure numbers under figures 1 and 3 and 4 in Figure 2.

**AC:** As Figure 2 was explicitly asked for by the reviewers and serves as a graphical overview of our approach, we decided to keep it. Thank you very much for your helpful suggestions to improve the

figure. In revised Figure 2, we included the terms "Step 1" and "Step 2", and we added figure and section numbers.

**EC:** Line 202 - It is not clear what the authors mean by 'Figure 2, left box' (is it the blue box)? If the authors clearly label each box as Step 1, Step 2, etc. then this confusion is reduced. This is especially important because the authors use these terms in the text when referring to this figure.

**AC:** We replaced the terms "right box" and "left box" with "Step 1" and "Step 2" in lines 192, 193, 200, and 201.

**EC:** There were a total of 5 workshops and 2 focus groups. To increase readability, I suggest creating a table that lists workshop names (e.g., workshop #1) together with other relevant information such as the time of each workshop, format/structure, participants' backgrounds, workshop aims, activities, and evaluation strategies. This information when structured in a table can be easier to read and digest.

**AC:** The manuscript only refers to what we have done before and during the first workshop and an evaluation after the whole participatory process. We only mention the other workshops and the focus groups to highlight that the quantification and communication presented in our study laid the basis for the following workshops. This is why we refrained from including a table with more details on the workshops and focus groups.

**EC:** Spell our RCP the first time it appears in the text (Figure 1). Follow the same rule for other abbreviations used in the text.

**AC:** We included "representative concentration pathways (RCP)" in line 174 when the abbreviation "RCP" was used the first time, which is in a figure heading. This is why we also spelled it out in lines 239-240 when the term was used the first time in the text. In line 143, we included "global climate models (GCM)".

Other changes

**AC:** We decided to use the term "study" consistently throughout the text (instead of "paper"). Moreover, we included a thank you to the editors in the Acknowledgements and corrected spelling errors. However, we again did not follow the remarks from the precedent review file validation because Figure 6 was presented in this way to the stakeholders in the first workshop. But we discuss in lines 561-563 that the colors in Figure 6 should be changed to colors from color palettes visible for persons with color vision deficiency.